# Histone methyltransferase DOT1L coordinates AR and MYC stability in prostate cancer

R. Vatapalli[1], V. Sagar[1], Y. Rodriguez[1], J. C. Zhao[2], K. Unno[1], S. Pamarthy[3], B. Lysy[1], J. Anker [1], H. Han[1], Y. A. Yoo[1], M. Truica[1], Z. R. Chalmers[1], F. Giles[4], J. Yu [2], D. Chakravarti [5,6], B. Carneiro[7] & S. A. Abdulkadir [1,6,8 ✉]

The histone methyltransferase DOT1L methylates lysine 79 (K79) on histone H3 and is involved in Mixed Lineage Leukemia (MLL) fusion leukemogenesis; however, its role in prostate cancer (PCa) is undefined. Here we show that *DOT1L* is overexpressed in PCa and is associated with poor outcome. Genetic and chemical inhibition of DOT1L selectively impaired the viability of androgen receptor (AR)-positive PCa cells and organoids, including castration-resistant and enzalutamide-resistant cells. The sensitivity of AR-positive cells is due to a distal K79 methylation-marked enhancer in the *MYC* gene bound by AR and DOT1L not present in AR-negative cells. DOT1L inhibition leads to reduced MYC expression and upregulation of MYC-regulated E3 ubiquitin ligases HECTD4 and MYCBP2, which promote AR and MYC degradation. This leads to further repression of MYC in a negative feed forward manner. Thus DOT1L selectively regulates the tumorigenicity of AR-positive prostate cancer cells and is a promising therapeutic target for PCa.

[1] Department of Urology, Northwestern University Feinberg School of Medicine, Chicago, IL, USA. [2] Division of Hematology/Oncology, Department of Medicine, Northwestern University Feinberg School of Medicine, Chicago, IL, USA. [3] Atrin Pharmaceuticals, Pennsylvania Biotechnology Center, Doylestown, PA, USA. [4] Developmental Therapeutics Consortium, Chicago, IL, USA. [5] Division of Reproductive Science in Medicine, Department of OB/GYN, Northwestern University Feinberg School of Medicine, Chicago, IL, USA. [6] The Robert H. Lurie Comprehensive Cancer Center, Northwestern University Feinberg School of Medicine, Chicago, IL, USA. [7] Lifespan Cancer Institute, Division of Hematology/Oncology, Alpert Medical School, Brown University, Providence, RI, USA. [8] Department of Pathology, Northwestern University Feinberg School of Medicine, Chicago, IL, USA. ✉email: Sarki. abdulkadir@northwestern.edu

Histone methyltransferases have emerged as important therapeutic targets in oncology but there is limited knowledge about their contributions to the pathogenesis of several malignancies[1]. Disruptor of Telomeric silencing 1 Like (DOT1L) is a histone methyltransferase that methylates Lysine 79 of histone H3[2]. H3K79 methylation is mainly associated with active transcription, transcription elongation, and DNA repair response[3–8]. Previous studies have uncovered an important role for DOT1L in driving pathogenesis of acute myeloid leukemias (AML) with mixed lineage leukemia (MLL) gene translocations[4,9]. Other studies have demonstrated the efficacy of DOT1L inhibition in solid tumors[10–13], however its role in prostate cancer (PCa) is yet to be delineated. PCa is the most common adult malignancy in men and the second most lethal[14]. The mainstay treatment for advanced PCa involves targeting of the androgen receptor (AR) signaling pathway[15]. Although most patients initially respond to treatment, many progress to develop Castration-Resistant Prostate Cancer (CRPC)[14,16]. The main oncogenic driver of CRPC is sustained signaling by the AR[17–21]. While newer AR targeting therapies like Enzalutamide (ENZA) have improved survival of patients, resistance frequently occurs through various mechanisms including persistent activation of the AR pathway[19,22–24].

In addition to AR, deregulation of c-MYC has been observed in over 60% of CRPC patients[25–29]. Sustained MYC expression is required for the viability of CRPC cells[27] and crosstalk between MYC and AR at the level of target gene expression has been described[25]. While MYC is recognized as a valued therapeutic target in CRPC and recent studies have identified new promising direct MYC inhibitors[30], there is significant interest in strategies targeting MYC by indirect mechanisms[31]. In this study, we demonstrate that DOT1L inhibition impairs PCa tumorigenicity by concurrently suppressing AR and MYC proteins.

## Results

### High DOT1L expression correlates with poor outcome in prostate cancer.
We investigated a potential role for DOT1L in PCa by screening for DOT1L alterations in several PCa datasets. The analysis revealed that *DOT1L* expression was significantly upregulated in PCa relative to normal prostate (Fig. 1a and Supplementary Fig. 1a). We confirmed these results in an independent validation set of benign and PCa patient samples (Fig. 1c). We also found *DOT1L* overexpression in in other solid cancer types including breast cancer, glioblastomas relative to normal tissues (Supplementary Fig. 1b–g). We next examined the correlation of *DOT1L* overexpression to tumor grade and outcome in PCa patients. *DOT1L* expression was associated with Gleason score (Supplementary Fig. 1h). In addition, high *DOT1L* expression was significantly associated with poor disease-free survival (Fig. 1b and Supplementary Fig. 1i) in multiple datasets. Moreover, in patients with cancers of intermediate grade (Gleason 7), high *DOT1L* expression was still able to significantly predict poor disease-free survival (Fig. 1b, right panel). DOT1L is the only enzyme known to catalyze H3K79 methylation, so we checked the levels of DOT1L-mediated H3K79 methylation in patient tissues. Immunohistochemistry (IHC) for H3K79me2 in a tissue microarray (TMA) with 80 PCa and 80 benign prostate specimens showed increased H3K79me2 staining in PCa relative to benign tissues (Fig. 1d).

### DOT1L is required for viability of AR-positive PCa cells.
To ascertain the functional role of DOT1L in PCa, we treated a panel of PCa cell lines with specific DOT1L inhibitor EPZ004777 (EPZ) and performed colony formation and cell viability assays. DOT1L inhibition led to a selective loss in colony formation and cell viability in AR-positive cells compared to AR-negative cells, indicating that response to DOT1L inhibition may depend on AR signaling status (Fig. 2a, b). AR-positive CRPC cells (C42B), AR variant AR-V7 expressing cells (22Rv1) and ENZA resistant cells were all sensitive to DOT1L inhibitor (Fig. 2a–c; Supplementary Fig. 2a). A second DOT1L inhibitor, EPZ5676, showed similar results as EPZ004777 (Fig. 2d). shRNA knockdown of DOT1L in LNCaP cells also decreased colony formation (Fig. 2e, Supplementary Fig. 2b). To further demonstrate that DOT1L inhibition is effective in AR positive PCa, we used a patient-derived xenograft (PDX) human CRPC model, TM00298. PDX organoids treated with EPZ004777 and EPZ5676 showed a substantial loss in cell viability (Fig. 2f). Transduction of PDX organoids with shDOT1L retrovirus also led to a dramatic loss in organoid viability compared to shControl retrovirus (Fig. 2f). Similar to 2D cultures, LNCaP organoids but not AR-negative PC3 organoids were sensitive to EPZ (Supplementary Fig. 2c). Next, we sought to test the effects of DOT1L inhibition in vivo. Due to the poor pharmacokinetic properties of the compounds, we treated LNCaP cells with EPZ for 7 days, then inoculated viable cells into mice subcutaneously. In vivo, the growth of EPZ pretreated tumors was substantially inhibited compared to the control group (Fig. 2g). These results indicate that DOT1L inhibition has sustained effects on the PCa cells, possibly by remodeling of the epigenetic landscape.

Due to the known long half-life of H3K79 methylation in cells[32,33], we performed the above experiments after 8–12 days of inhibitor treatment or shRNA expression. Short-term treatment with EPZ (Supplementary Fig. 2d) or transient knockdown of DOT1L with siRNA had no effect on the cell viability of sensitive cell lines (Supplementary Fig. 2f). These data suggest that the effect of DOT1L inhibition in AR-positive cells is dependent on loss of H3K79 methylation. This was supported by the observation that H3K79me2 was decreased only after long-term (8d) treatment with EPZ but not short-term treatment (Fig. 2h–k and Supplementary Fig. 2e). These results support a model wherein the functional effects of DOT1L inhibition in the AR-positive cells require events that occur after the loss of H3K79 methylation such as dysregulated target gene expression in contrast to direct effects, such as modification of AR by DOT1L enzymatic activity or protein–protein interaction.

The differential sensitivity to DOT1L inhibition between AR-positive and AR-negative cells was not due to lack of inhibition of DOT1L function in the AR-negative cells as 8 days treatment with EPZ decreased H3K79me2 to the same extent in both LNCaP and PC3 cells (Fig. 2h, i). This was confirmed with the shDOT1L construct and EPZ5676 (Fig. 2j). In addition, there was no difference in the baseline expression of *DOT1L* or H3K79me2 protein levels in sensitive versus resistant cell lines (Supplementary Fig. 2g–h).

We next sought to determine if differential distribution of H3K79 methylation at baseline or after DOT1L inhibition may be related to sensitivity to EPZ. To this end, we performed Chromatin Immunoprecipitation-sequencing (ChIP-seq) for H3K79me2 in both LNCaP and PC3 cells. EPZ decreased the number of H3K79me2 marked peaks to the same extent (Fig. 2l); however, while a subset of sites was shared between the two cell lines, the majority of peaks were unique to each cell line (Supplementary Fig. 2i). To characterize the unique H3K79me2 marked genes in each cell line, we performed ChIP Enrichment Analysis (ChEA). We found that these genes were associated with distinct transcription factors that function in each cell. In LNCaP, AR and FOXA1 associated genes were among the top hits (Supplementary Fig. 2j), while in PC3, top hits included neural lineage associated transcription factors HOXC9 and MYCN (Supplementary Fig. 2j). We also compared the sensitivity of

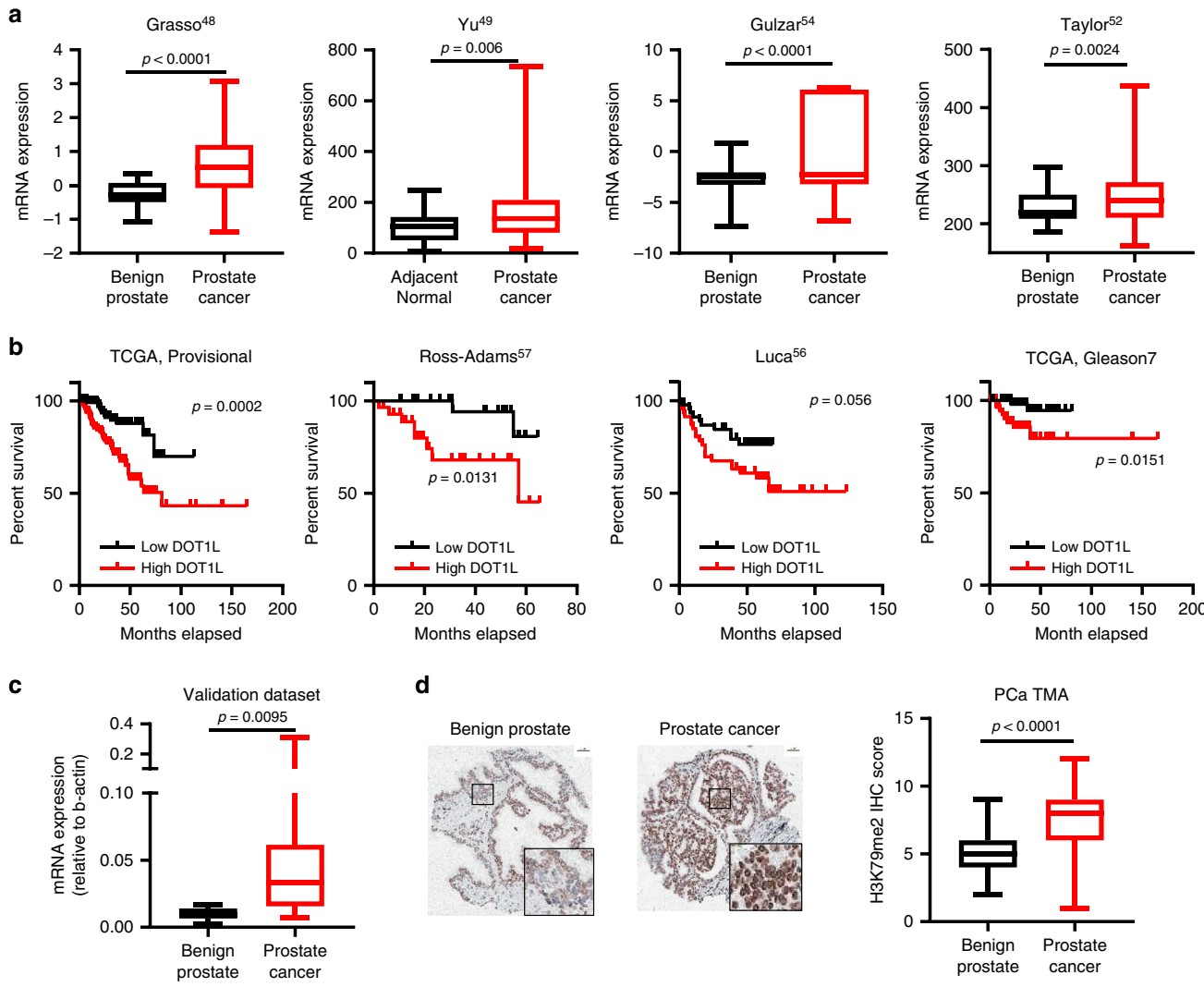

**Fig. 1 DOT1L expression is upregulated in prostate cancers and high expression correlates with poor disease-free survival. a** Comparison of *DOT1L* expression in four cohorts of prostate cancer patients' datasets. Data for this analysis were used from the Grasso PCa dataset [benign (n = 28), PCa (n = 94)], Yu PCa dataset [normal (n = 60), PCa (n = 81)], Gulzar Prostate cancer dataset [normal (n = 66), prostate cancers (n = 83) and Taylor PCa dataset [normal (n = 29), PCa (n = 150)]. **b** Disease-free Survival analysis of three independent cohorts of prostate cancer patients divided by expression of *DOT1L* using 25 percentile and 75 percentile cutoffs. Data was used from the Luca CancerMap prostate cancer dataset [n = 46 per group], TCGA prostate cancer dataset [n = 122 per group], Ross-Adams Discovery dataset [n = 27 per group] and TCGA Gleason 7 patients [n = 61 per group]. **c** Comparison of DOT1L expression in an independent dataset with PCa patient specimens [benign (n = 15), PCa (n = 45)]. **d** Representative images (left) and comparison of average H3K79me2 staining scores in tissue sections from a PCa TMA [normal (n = 80), PCa (n = 80)] Scale bar indicates 50 µM. Statistical tests: p value determined by two-sided Welsh's t test (**a**, **c**, **d**) and Log-rank test. For box plots, minima and maxima values are shown (**a**, **c**, **d**).

H3K79me2 marked genic regions versus intergenic regions in both cell lines and found that while the H3K79me2 genic regions were equally affected in LNCaP cells and PC3 cells, the intergenic regions were specifically sensitive to EPZ depletion in the LNCaP cells (Supplementary Fig. 2k).

**DOT1L inhibition leads to impairment of the AR pathway**. Since AR-positive cell lines were sensitive to DOT1L loss, we assessed the status of the AR pathway after short-term and long-term DOT1L inhibition. AR protein levels were decreased upon long-term treatment with EPZ in a dose dependent manner and upon DOT1L knockdown in PCa cell lines and PDX organoids (Fig. 3a–d). Notably, AR protein was unaffected with short-term DOT1L inhibition (Supplementary Fig. 3a). Conversely, we found that in DOT1L overexpressing LNCaP cells (LNCaP-DOT1L), AR protein levels were upregulated (Fig. 3e). Moreover, LNCaP-DOT1L cells displayed an increased proliferation rate in charcoal

stripped medium devoid of androgens (Fig. 3f) suggesting that DOT1L promotes androgen independent growth of PCa cells.

DOT1L does not regulate AR at the transcriptional level as we found no change in *AR* mRNA levels after EPZ treatment or DOT1L knockdown (Supplementary Fig. 3b). We therefore examined the possibility that AR protein stability was altered upon DOT1L loss. Indeed, AR protein half-life was reduced in LNCaP cells treated with EPZ when compared to DMSO treated cells in a cycloheximide chase assay (Fig. 3g).

To examine the status of the downstream AR pathway upon DOT1L inhibition, we first assessed the levels of *PSA*, a canonical AR target gene. PSA protein levels were decreased in a dose dependent manner upon EPZ treatment (Fig. 3h). In addition, using an Androgen Responsive Element (ARE) promoter-GFP reporter expressing cell line to monitor AR pathway activation, we found a dose dependent decrease in AR transcriptional activity with long-term EPZ treatment (Fig. 3i). We then

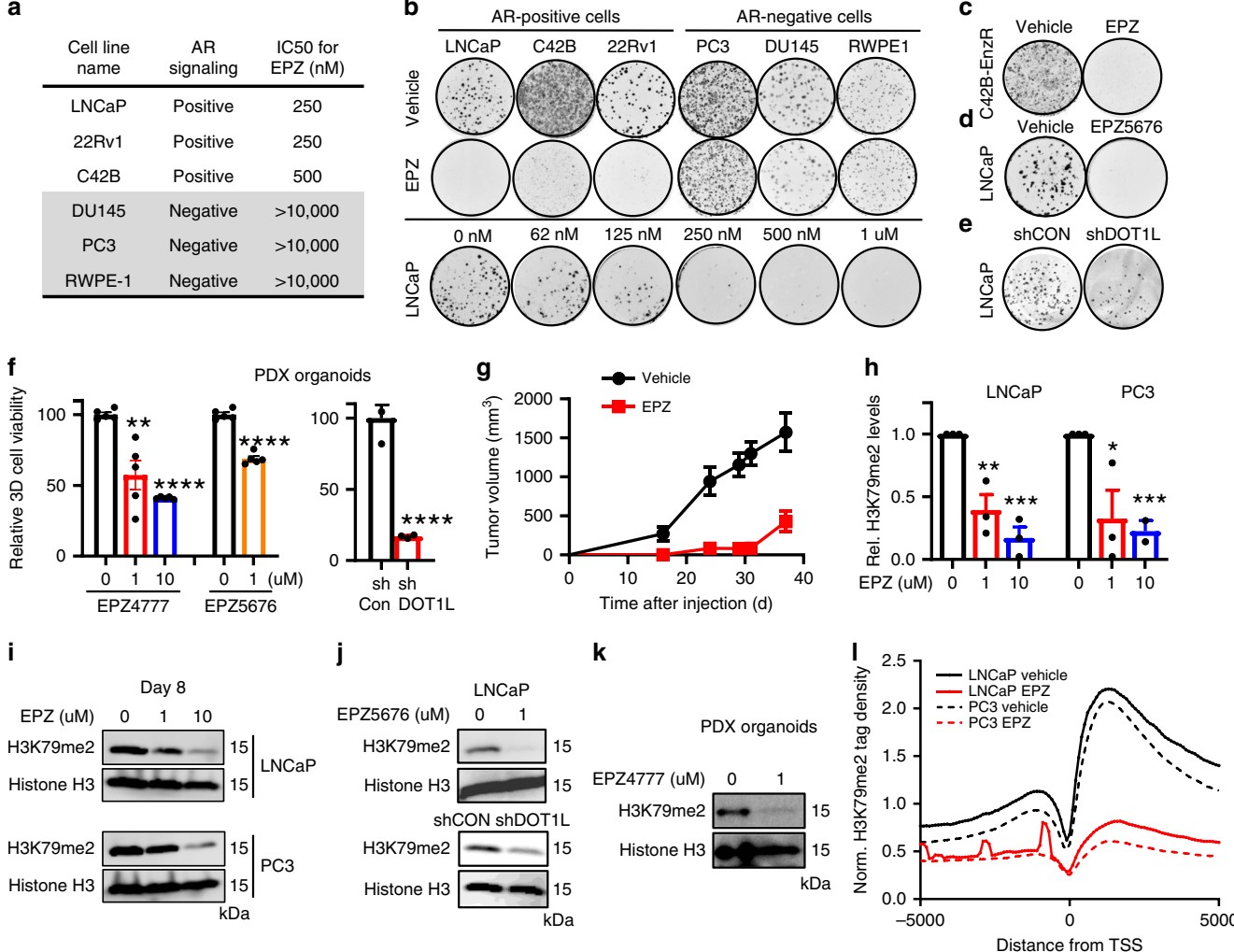

**Fig. 2 DOT1L inhibition leads to selective loss in cell viability in AR-positive cells. a** Comparison of Half maximum inhibitory concentration (IC50) for EPZ in 6 prostate cancer cell lines. **b** (Top) Representative images of clonogenic assays performed for 6 prostate cancer cell lines treated with Vehicle or 10 μM of EPZ for 12 days from 3 independent experiments. (Bottom) Representative images of clonogenic assay in LNCaP cells treated with increasing doses of EPZ for 12 days from 3 independent experiments. Colony formation assays performed in (**c**) C42B-ENZR cells treated with Vehicle or 10 μM EPZ for 12 days. (**d**) LNCaP cells treated with Vehicle or 10 μM EPZ5676 for 12 days. **e** LNCaP cells transduced with shControl or shDOT1L followed by clonogenic assays evaluated after 12 days. **f** PDX organoids were treated with Vehicle, EPZ (left) or shControl or shDOT1L (middle) and Vehicle or EPZ5676 (right). **g** LNCaP cells were treated in vitro with Vehicle or 10 μM EPZ and 2 million viable cells were injected subcutaneously in NOD-SCID mice and tumor growth was monitored (n = 5 per arm). **h, i** H3K79me2 western blot analysis performed after 8 days of treatment with Vehicle or EPZ with quantitation of protein levels. **j** H3K79me2 western blot analysis in LNCaP after 8 days of treatment with (top) Vehicle or EPZ5676, (bottom) shControl or shDOT1L (Representative experiment shown). **k** H3K79me2 western blot analysis in PDX organoids after 8 days of treatment with Vehicle or 1 μM EPZ. **l** Histogram of H3K79me2 tags (within 10 kb) centered on the TSS in LNCaP and PC3 Vehicle and EPZ-treated samples. ChIP-seq performed after cells were treated for 8 days with Vehicle or 1 μM EPZ. Statistical tests: p value determined by two-tailed Student's t test (**f–h**). n = 5 (**f, g**) and n = 3 (**h–j**) independent experiments. Error bars represent S.E.M. *p < 0.05; **p < 0.01; ***p < 0.001; ****p < 0.0001.

performed genome-wide expression profiling and interrogated the AR transcriptional program. Surprisingly, while a small subset of AR activated genes was suppressed by EPZ treatment as expected (e.g., *PSA (KLK3), TMPRSS2, KLK2*), a larger subset of AR targets in the Nelson_Response_To_Androgen_Up dataset was paradoxically upregulated, including *ELL2, HERC3*, and *ACSL3* (Fig. 3j and Supplementary Fig. 3h). We confirmed these results using qRT-PCR after both EPZ treatment and shDOT1L expression (Fig. 3k, l). We also confirmed that AR binding at these target gene promoters was decreased upon EPZ treatment by ChIP (Fig. 3m). These results were even more surprising considering the fact that multiple androgen metabolism gene sets were upregulated upon EPZ treatment (Supplementary Fig. 3c–e). We observed an upregulation of members of the UGT2B family

of enzymes that are responsible for androgen glucuronidation leading to removal of androgens from the cell[34,35]. UGT2B7, 15 and 17 are the main enzymes involved in the process[34,35] and these were upregulated consistently in LNCaP and C42B cells upon EPZ treatment (Supplementary Fig. 3e–f). Since these genes are negatively regulated by androgen bound AR[36], we confirmed that AR enrichment at *UGT2B15* and *UGT2B17* promoters was decreased upon EPZ treatment (Supplementary Fig. 3g). Overall, these results indicate that, in addition to impairing AR protein stability, DOT1L inhibition may also lead to loss of androgen levels in PCa cells by upregulating the UGT2B family of enzymes. All of these changes should lead to a reduction in AR transcriptional activity; yet paradoxically, we observed upregulation of a subset of AR activated genes as described above. This

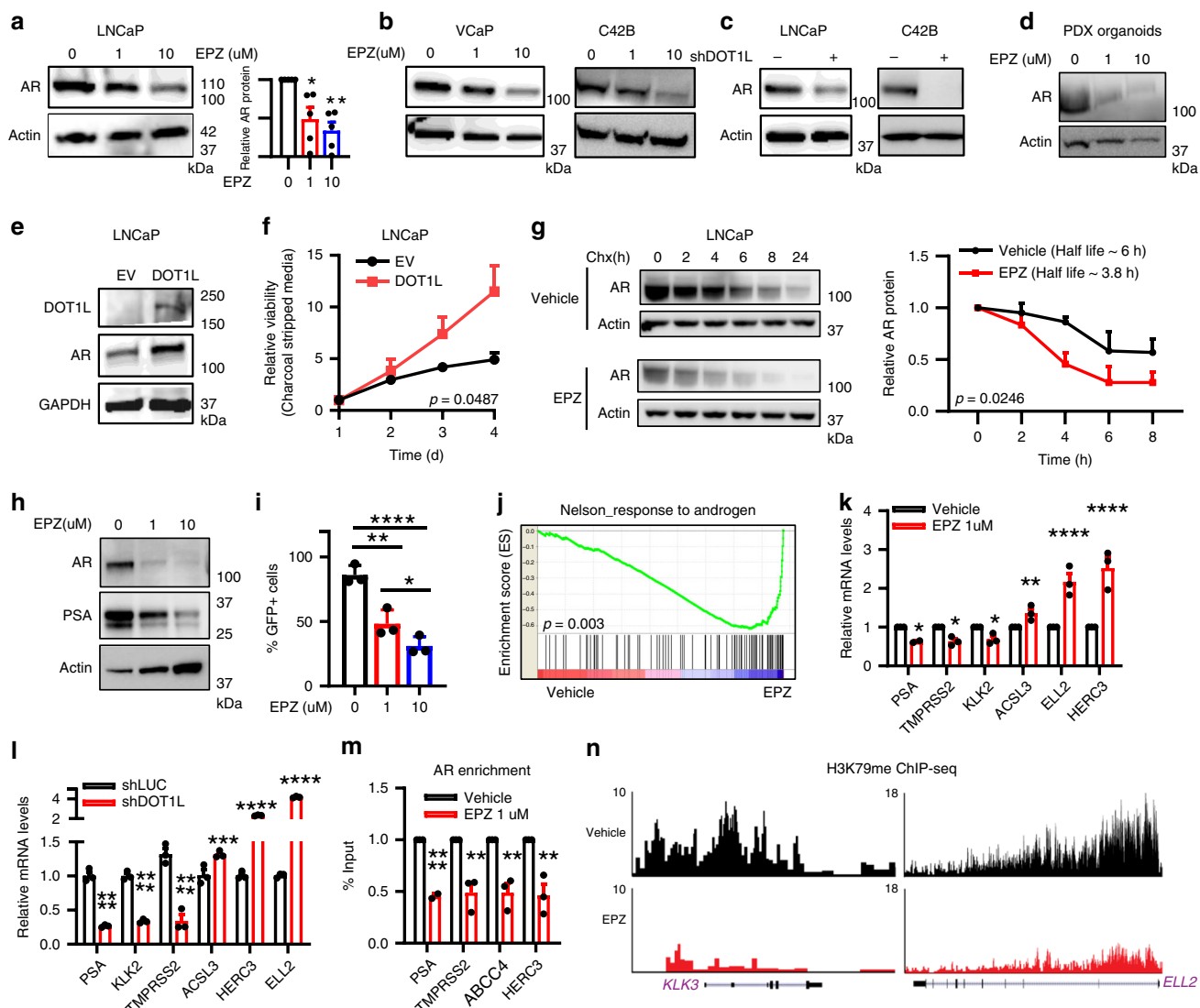

**Fig. 3 DOT1L inhibition leads to loss in AR protein levels by decreasing its protein stability.** AR protein levels in (**a**) LNCaP cells treated with EPZ followed by quantitation (*n* = 5). **b** AR protein in VCaP and C42B cells after EPZ treatment. **c** shControl or shDOT1L transfected LNCaP and C42B cells. **d** PDX organoids treated with EPZ. **e** AR protein levels in EV or DOT1L overexpressing LNCaP cells. **f** Proliferation assay of LNCaP cells with and without DOT1L expression in charcoal stripped media measured using MTS assays for 4 days. **g** AR western analysis after 50 μg/ml Cycloheximide treatment in LNCaP cells treated with Vehicle or EPZ for 8 days (left). Representative experiment shown. Quantitation of AR protein levels from 3 independent experiments (right). **h** PSA and AR western blot analysis in LNCaP treated with Vehicle and EPZ for 8 days. Representative image shown. **i** Percentage of RFP + GFP + cells counted by Flow cytometry after 8 days of Vehicle or EPZ treatment in LNCaP cells transfected with ARE-GFP reporter construct. **j** GSEA plot of Nelson_Response_to_Androgen geneset enriched in LNCaP cells treated with 1 μM EPZ for 8 days compared to Vehicle treatment. mRNA expression of 6 AR target genes measured by qRT-PCR in LNCaP cells after 8 days of (**k**) 1 μM EPZ treatment and (**l**) transduction with shDOT1L or shControl lentivirus. **m** Relative enrichment of AR at 4 target genes measured by ChIP followed by qPCR in LNCaP cells treated with Vehicle or 1 μM EPZ for 8 days. **n** ChIP-seq plots of H3K79me2 at two AR target genes in LNCaP cells treated with Vehicle or 1 μM EPZ. Statistical tests: *p* value determined by two-tailed *t* test (**a**, **f–i**, **k–m**) corrected for multiple comparisons in (**k–m**). *n* = 3 independent experiments (**b**, **c**, **f**, **h–m**). FDR < 25% (**j**). Error bars represent S.E.M. \**p* < 0.05; \*\**p* < 0.01; \*\*\**p* < 0.001; \*\*\*\**p* < 0.0001.

discrepancy is not due to persistence of H3K79me2 levels at these upregulated AR target genes as determined by inspection of H3K79me2 ChIP-seq plots (Fig. 3n) and by ChIP-qPCR (Supplementary Fig. 3i). These observations led us to hypothesize that upregulation of these discordant AR target genes may be due to another transcription factor that is modulated by DOT1L inhibition.

**DOT1L inhibition suppresses the MYC pathway.** To search for a candidate DOT1L inhibitor-regulated factor that may cross-regulate expression of the discordant AR target genes we examined EPZ-treated LNCaP and PC3 cells for significant transcription factor regulated gene Geneset enrichment analysis (GSEA) datasets. We found alterations in the MYC pathway with MYC target gene sets suppressed in EPZ-treated LNCaP cells. Importantly, MYC has previously been shown to repress a subset of AR target genes in PCa by Barfeld et al.[25]. We therefore hypothesized that the discordant AR target genes we observed after EPZ treatment may be co-regulated by MYC, with loss of MYC induced by EPZ treatment leading to their upregulation, despite the reduction in AR levels. To examine this notion, we

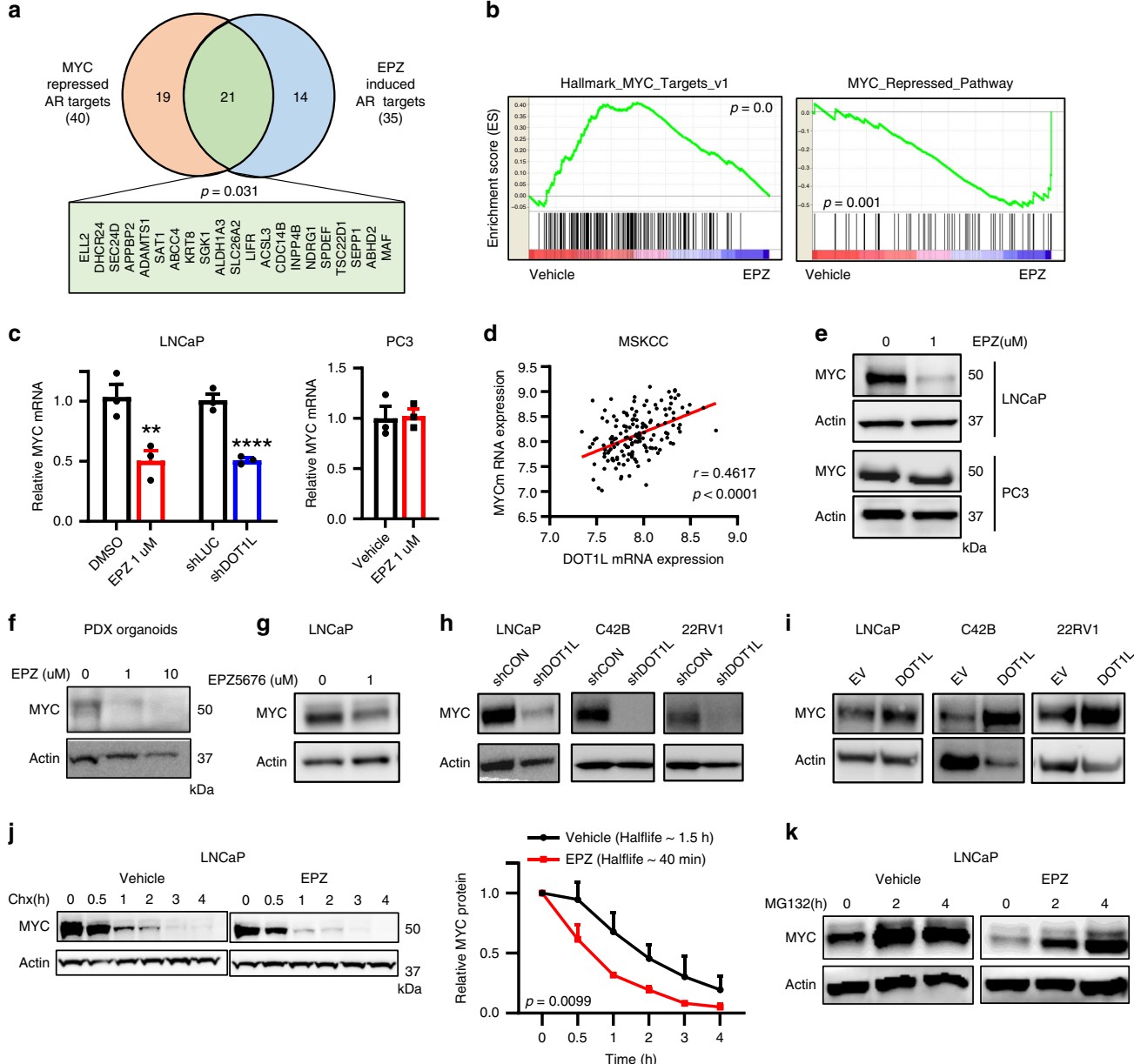

**Fig. 4 DOT1L inhibition leads to impairment of MYC pathway. a** Comparison of leading-edge genes from Nelson_Response_To_Androgen dataset in the present study (n = 35) and Barfeld study (n = 40). **b** GSEA plots of two MYC related datasets enriched in either Vehicle treated (left) or EPZ-treated (right) LNCaP cells. Cells were treated with 1 μM Vehicle or EPZ for 8 days prior to microarray analysis (n = 3). **c** mRNA expression of *MYC* in (left) LNCaP cells treated with Vehicle or 1 μM EPZ and LNCaP cells transduced with shControl or shDOT1L lentivirus; (right) PC3 cells treated with Vehicle or 1 μM EPZ for 8 days. **d** Positive correlation between *DOT1L* and *MYC* expression in the MSKCC dataset (n = 150). Data were obtained from cbioportal.org. **e** Western blot analysis of MYC protein in LNCaP and PC3 cells treated with Vehicle or EPZ 1 μM for 8 days. Representative images shown. Western blot analysis of MYC protein in (**f**) PDX organoids treated with EPZ for 8 days, **g** LNCaP cells treated with EPZ5676 for 8 days, **h** LNCaP, C42B, and 22rv1 cells with DOT1L knockdown and **i** LNCaP, C42B, and 22rv1 cells with DOT1L overexpression. Representative images shown. **j** Western blot analysis and quantitation of MYC protein after treatment with 50 ug/ml Cycloheximide in LNCaP cells treated with vehicle or 1 μM EPZ for 8 days. **k** Western blot analysis of MYC protein after treatment with 10 μM MG-132 in LNCaP cells treated with vehicle or 1 μM EPZ for 8 days. Representative images shown. Statistical tests: p value determined by Hypergeometric test (**a**), Spearman's rank correlation (**d**), and two-tailed Student's t test (**c**, **j**). Error bars represent S.E.M. n = 3 independent experiments (**c**, **e**–**k**). **p < 0.01; ****p < 0.0001.

first compared the leading-edge genes from the MYC repressed/ AR targets defined by Barfeld with the EPZ-induced AR targets leading-edge genes from our own study and found a significant overlap (p value = 0.03) (Fig. 4a).

Examination of gene expression profiling data by GSEA showed that multiple MYC induced gene sets were repressed upon DOT1L inhibition in LNCaP cells but not PC3 cells (Fig. 4b and Supplementary Fig. 4a). Although one MYC related geneset

(Schlosser_MYC_targets_ repressed_by_serum) appeared to be repressed in PC3 cells, further analysis of the leading-edge genes revealed that they could be altered upon perturbations in other transcription factors as well (Supplementary Fig. 4b). Hence, these data indicate that EPZ treatment suppresses the MYC pathway in LNCaP but not PC3 cells. We found that *MYC* mRNA levels were reduced in LNCaP and C42B cells but not PC3 cells after EPZ treatment (Fig. 4c and Supplementary Fig. 4d).

Furthermore, *MYC* expression positively correlated with *DOT1L* expression in multiple patient datasets (Fig. 4d and Supplementary Fig. 4e).

We next analyzed MYC protein levels in PCa cell lines and organoids after DOT1L inhibition. MYC protein levels were dramatically decreased by EPZ treatment in AR-positive cells LNCaP, C42B, 22rv1 and the AR-positive PDX organoids but not in PC3 cells (Fig. 4e, f). We confirmed these results using EPZ5676 and DOT1L shRNA (Fig. 4g, h). Notably, MYC protein levels were consistently decreased only after long-term EPZ treatment starting at 8 days in LNCaP cells (Supplementary Fig. 4c), suggesting that full reduction in MYC protein is dependent on the loss of H3K79 methylation. Conversely, overexpression of DOT1L in LNCaP, C42B and 22rv1 cells led to a significant upregulation in MYC levels (Fig. 4i). By the cycloheximide chase assay, we found decreased MYC protein half-life in EPZ-treated LNCaP but not PC3 cells (Fig. 4j and Supplementary Fig. 4g). We observed no changes in levels of MYC phosphorylated at Threonine-58 or Serine-62 indicating that changes in these post-translational modifications that are known to regulate MYC stability are not affected by DOT1L inhibition (Supplementary Fig. 4f). Treatment with the proteasome inhibitor, MG-132 restored MYC protein levels in EPZ-treated LNCaP cells, implying that MYC is degraded through the proteasomal pathway upon EPZ treatment (Fig. 4k). These data suggest that DOT1L dependent MYC loss is dependent on both DOT1L mediated H3K79 methylation and AR activity.

**DOT1L inhibitor-regulated E3 ligases target AR and MYC stability.** Since both AR and MYC proteins were degraded at increased rates upon long-term DOT1L inhibition, we hypothesized that DOT1L inhibition and loss of H3K79 methylation might impact the expression of E3 ligases that regulate the stability of MYC and AR. A search of EPZ-regulated genes in LNCaP cells for known/putative E3 ubiquitin ligases identified four candidates: *HERC3, HECTD4, MYCBP2,* and *TRIM49* (Fig. 5a). The expression levels of *HERC3, HECTD4,* and *MYCBP2* were increased upon DOT1L inhibition, consistent with these ligases playing a role in AR/MYC degradation. *TRIM49* expression on the other hand was decreased after DOT1L inhibition, which will not be consistent with this protein promoting AR/MYC degradation. However, some E3 ligase family members can stabilize proteins. For example, TRIM39 has been shown to stabilize MOAP-1 and Cactin by modulating polyubiquitination[37,38]. We confirmed consistent dysregulation of all four genes by EPZ in LNCaP but not PC3 cells (Fig. 5b and Supplementary Fig. 5a). Similar results were obtained with DOT1L knockdown (Supplementary Fig. 5b). Overexpression of DOT1L decreased the expression of *HECTD4* and *MYCBP2* in LNCaP, C42B, and 22rv1 cells (Supplementary Fig. 5c), suggesting that DOT1L plays a role in repressing these targets either directly or indirectly.

We hypothesized that dysregulation of one or more of these candidate E3 ligases may mediate AR and MYC protein degradation. Therefore, we performed a siRNA knockdown screen of the 4 candidate ligases and evaluated AR and MYC protein levels (Fig. 5c; Supplementary Fig. 5d–g). We did not detect consistent changes in AR or MYC protein levels after HERC3 knockdown. However, knockdown of HECTD4 and MYCBP2 led to an increase in both AR and MYC proteins while TRIM49 depletion reduced AR and MYC levels (Fig. 5c and Supplementary Fig. 5d–g), consistent with their proposed roles after DOT1L inhibition. We did not observe any changes in AR and MYC mRNA levels after knockdown of HECTD4, MYCBP2, or TRIM49 supporting post-transcriptional regulation of AR and MYC by these ligases (Supplementary Fig. 5e–g). We also

overexpressed HECTD4 and MYCBP2 in LNCaP cells and saw a dramatic decrease in cell viability, similar to what is observed with DOT1L inhibition (Supplementary Fig. 5h). A simultaneous decrease in AR and MYC levels upon HECTD4 overexpression indicates its role as an AR and MYC targeting E3 ligase (Supplementary Fig. 5h). Altogether, these results suggest that DOT1L inhibitor-mediated upregulation of HECTD4 and MYCBP2 concomitant with downregulation of TRIM49 promote AR and MYC protein degradation.

To show this directly, we examined if knockdown of HECTD4 and MYCBP2 can rescue EPZ-mediated AR and MYC degradation. We therefore treated LNCaP cells with EPZ or vehicle for 6 days to allow loss of H3K79 methylation, then transfected cells with siHECTD4 and siMYCBP2 and analyzed AR and MYC protein levels 2 days later. Both AR and MYC levels were rescued after dual knockdown of HECTD4 and MYCBP2 in the EPZ-treated cells (Fig. 5d). These data strongly suggest that HECTD4 and MYCBP2 are primarily responsible for the EPZ-mediated loss of stability of AR and MYC proteins. To assess if the dual knockdown of HECTD4 and MYCBP2 could rescue the inhibitory effects of EPZ on cell viability, we repeated the same experiment, this time analyzing cell viability 6 days after siHECTD4 + siMYCBP2 transfection. The results indicate that HECTD4 and MYCBP2 depletion significantly rescued the EPZ treatment induced loss of cell viability (Fig. 5e). To examine the functional role of TRIM49 in regulating MYC and AR after EPZ treatment, we overexpressed TRIM49 in LNCaP cells treated with EPZ or vehicle for 8 days. Analysis of AR and MYC protein levels 2 days later showed that TRIM49 overexpression can partially restore the levels of AR & MYC after EPZ treatment (Supplementary Fig. 5i).

MYCBP2 is an established E3 ubiquitin ligase that has been previously shown to interact with MYC[39–42] and we found that MYCBP2 expression increased levels of ubiquitin conjugated MYC (Supplementary Fig. 5j). In addition, mass spectrometry data for AR-interacting proteins identified MYCBP2 (Supplementary Data 1). While the interaction of MYCBP2 and MYC has been reported, a role for HECTD4 or MYCBP2 as ubiquitin ligases/interacting partners of AR have not been demonstrated. By employing co-immunoprecipitation assays in cells expressing Flag-AR and Halotag-HECTD4 or MYC-tag-MYCBP2, we observed robust interaction between AR and the ubiquitin ligases HECTD4 and MYCBP2 (Fig. 5f–h). Furthermore, HECTD4 and MYCBP2 expression each increased levels of ubiquitin conjugated AR protein (Fig. 5g, h). We also found that low levels of *MYCBP2* or high levels of *TRIM49* correlated with poor disease-free survival in PCa patients (Supplementary Fig. 5k–l). In sum, these results show that DOT1L inhibition coordinates loss of AR and MYC protein stability by modulating the expression of multiple E3 ligases.

**MYC regulation of candidate E3 ligase expression.** DOT1L inhibition selectively dysregulates the expression of *HECTD4, MYCBP2,* and *TRIM49* E3 ligases in AR-positive but not AR-negative cells. To explore a direct role for AR in the regulation of these genes, we assessed their response to AR activation by DHT and AR inhibition by ENZA treatment (Supplementary Fig. 6a, b). *HECTD4* and *TRIM49* did not behave as AR target genes, while *MYCBP2* behaved like an AR stimulated target gene. Hence, these results did not recapitulate the expression profile seen upon EPZ treatment. We next examined the role of MYC in regulating the E3 ligases, including in the context of AR inhibition. Remarkably, depletion of MYC by siRNA in LNCaP cells resulted in upregulation of *HECTD4* and *MYCBP2* both in the presence and absence of AR inhibitor ENZA (Fig. 6a). *TRIM49* was upregulated

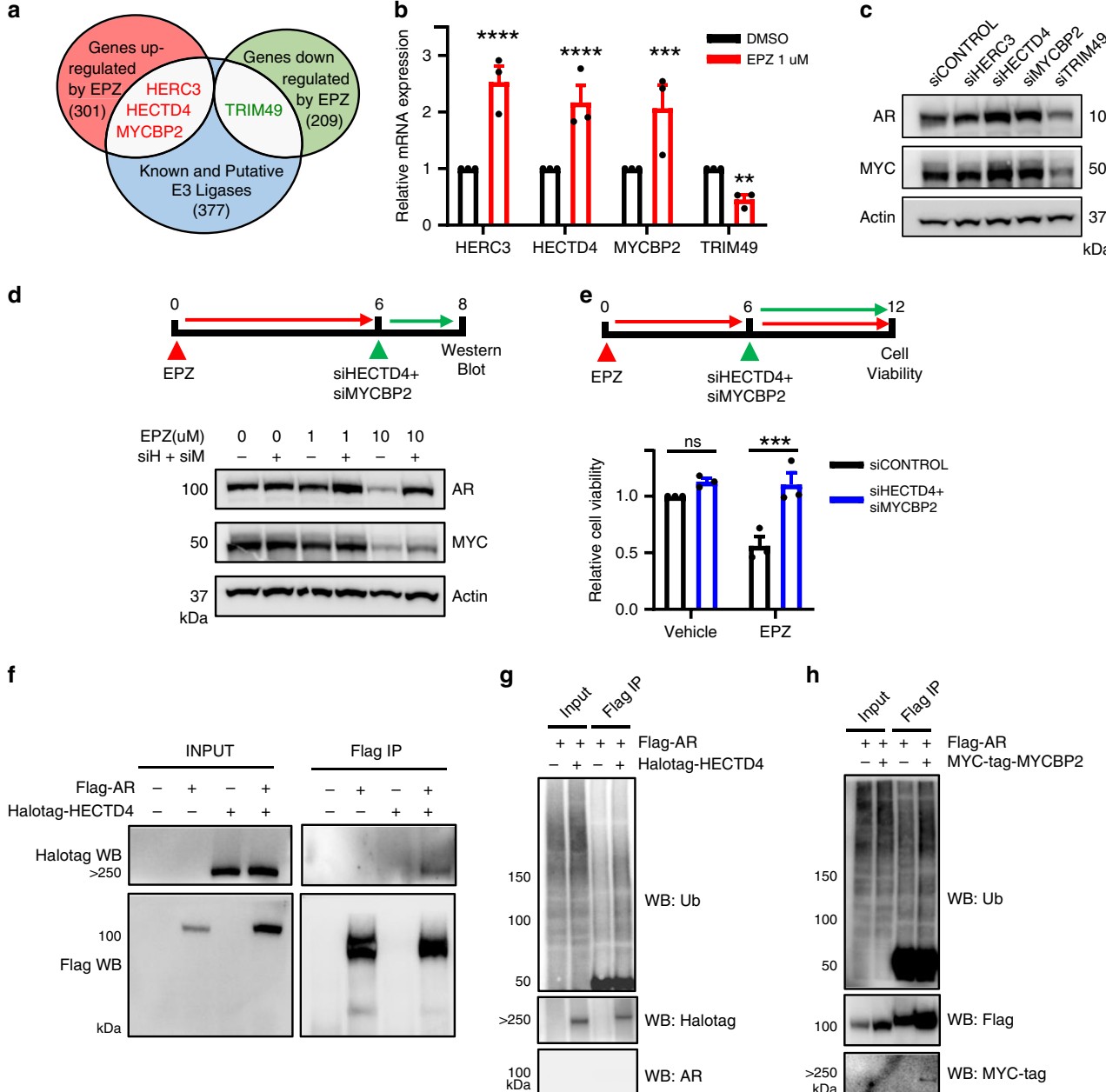

**Fig. 5 EPZ-regulated E3 ligases target AR and MYC stability. a** Comparison of genes that are differentially expressed by EPZ ($n = 510$) and known E3 Ubiquitin ligases. **b** mRNA expression of *HERC3, HECTD4, MYCBP2,* and *TRIM49* in LNCaP treated with vehicle or 1 μM EPZ for 8 days. **c** AR and MYC protein levels in LNCaP cells 2 days after being transfected by Control, HERC3, HECTD4, MYCBP2, and TRIM49 targeting siRNA. Representative images shown. **d** AR and MYC protein levels in LNCaP cells treated with Vehicle or EPZ for 6 days followed by transfection of Control or HECTD4 and MYCBP2 targeting siRNA for 2 days. Representative images shown. **e** Viability of LNCaP cells treated with Vehicle or EPZ for 6 days followed by transfection of Control or HECTD4 and MYCBP2 targeting siRNA measured by CCK8 assay at day 12. **f** Co-immunoprecipitation assays performed with Flag-AR pulldown followed by western blot analysis in 293T cells transfected with Flag-AR and Halotag-HECTD4 constructs after 2 days. **g** Flag-AR pulldown followed by Ubiquitin western analysis in 293T cells transfected with both Flag-AR and HECTD4 constructs. **h** Co-immunoprecipitation assays performed with Flag-AR pulldown followed by western blot analysis in 293T cells transfected with Flag-AR and MYC-tag-MYCBP2 constructs after 2 days. Statistical tests: *p* value determined by Student's *t* test (**b, e**). Error bars represent S.E.M. $n = 3$ independent experiments (**b–e**), $n = 2$ independent experiments (**g, h**). **$p < 0.01$; ***$p < 0.001$; ****$p < 0.0001$.

upon MYC knockdown but suppressed in the presence of ENZA (Supplementary Fig. 6c). The expression changes in *HECTD4, MYCBP2,* and *TRIM49* observed upon concomitant MYC and AR inhibition in these experiments are remarkably similar to the changes seen after EPZ treatment. In AR-negative PC3 cells, MYC knockdown led to upregulation of *HECTD4* and *MYCBP2* and downregulation of *TRIM49* (Fig. 6b and Supplementary

Fig. 6d). Hence, *HECTD4* and *MYCBP2* are potential MYC repressed targets. We assessed if *MYC* and *DOT1L* expression were correlated with the expression of *MYCBP2* and *HECTD4* in clinical datasets. However, we were unable to determine significant trends possibly due to the effects of other E3 ligase targets and the indirect nature of the regulation (Supplementary Fig. 6e).

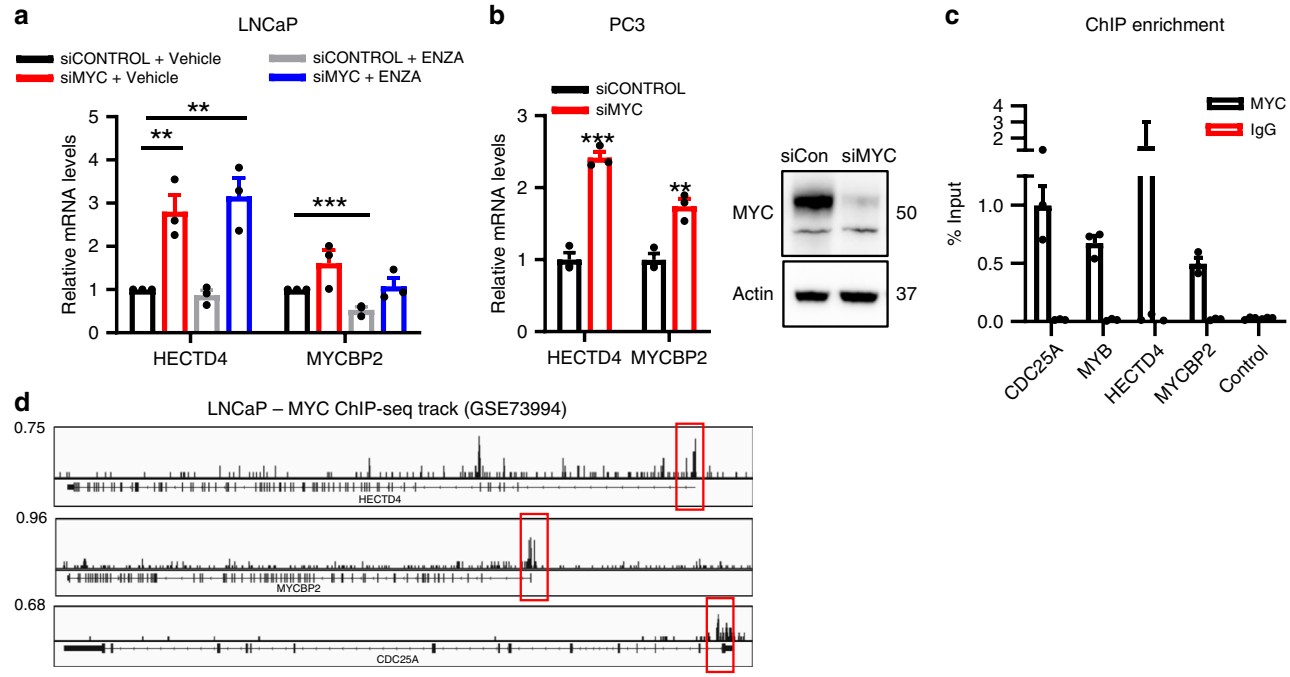

**Fig. 6 MYC loss induces expression of *HECTD4* and *MYCBP2*. a** mRNA expression of *HECTD4*, *MYCBP2* in LNCaP cells transfected with Control or MYC targeting siRNA (2 days) followed by treatment with Vehicle or 20 μM ENZA (2 days). **b** mRNA expression of *HECTD4*, *MYCBP2* and *TRIM49* in PC3 cells transfected with Control or MYC targeting siRNA for 2 days. **c** MYC ChIP followed by qPCR in LNCaP cells at the promoters of the indicated genes. **d** MYC ChIP-seq plots in LNCaP cells (GSE73994) at the indicated genes. Statistical tests: *p* value determined by two-tailed *t* test (**a**, **b**). Error bars represent S.E.M. *n* = 3 independent experiments (**a–c**). **\*\****p* < 0.01; **\*\*\****p* < 0.001.

To determine if MYC regulates the expression of these genes by directly binding to their promoters, we analyzed ChIP-seq data[25] and found MYC enrichment at both *HECTD4* and *MYCBP2* loci (Fig. 6d). We then validated MYC enrichment at these sites using ChIP-qPCR in LNCaP cells (Fig. 6c). These observations combined with the earlier results presented imply that EPZ-mediated loss of MYC expression triggers an increase in *MYCBP2* and *HECTD4* which then leads to loss of AR and MYC protein levels. Since this pathway seems to be restricted to AR expressing PCa cells, we asked the question—Why does DOT1L inhibition lead to decreased *MYC* expression in LNCaP cells but not in PC3 cells? A simple model to explain our observations is that DOT1L and AR co-regulate *MYC* gene expression in AR-positive cells.

**DOT1L and AR co-regulate MYC expression via a distal enhancer.** To determine the role of DOT1L and AR in regulating *MYC* expression we first examined the H3K79me2 landscape at the *MYC* gene locus before and after EPZ treatment. H3K79me2 levels across the *MYC* gene locus were uniformly reduced in both LNCaP and PC3 cells. Examination of AR binding at the *MYC* locus revealed an AR-bound enhancer 20 kb downstream of the *MYC* gene that has been proposed to regulate its expression[43]. Analysis of H3K79me2 ChIP-seq showed a broad H3K79me2 peak at this site in LNCaP cells (but not PC3 cells) that is lost after EPZ treatment (Fig. 7a and Supplementary Fig. 7a). Enhancer marks including H3K27ac overlapped with H3K79me2 and AR peaks at this site in LNCaP cells. Analysis of ChIP-seq data from three human PCa patient samples, showed AR and H3K27ac peaks at this enhancer (Fig. 7b). Furthermore, using ChIP-qPCR on PDX tumor tissue, we detected AR, DOT1L and H3K79me2 at this *MYC* enhancer in addition to the active enhancer marks H3K27ac and H3K4me2 and RNA polymerase II (Pol II) (Fig. 7c). Another more distal previously identified enhancer 1.7 Mb downstream of the *MYC* gene in murine leukemia cells and hematopoietic stem cells[44,45] showed no

enrichment of H3K79me2 or AR, indicating that it is not active in prostate cells (Supplementary Fig. 7b-c).

We hypothesized that AR binding to this H3K79me2 marked enhancer regulates MYC expression. This is supported by the observation that AR levels correlate with MYC expression in multiple datasets (Supplementary Fig. 8a). After DHT treatment, AR was recruited to the *MYC* enhancer in LNCaP and C42B cells and K79 methylation was increased (Fig. 7d and Supplementary Fig. 8b). Conversely, AR recruitment to the enhancer and MYC expression were reduced following ENZA treatment (Supplementary Fig. 8c). Moreover, DOT1L overexpression in LNCaP cells rescued the loss of MYC expression seen upon ENZA treatment (Supplementary Fig. 8d). We also found that AR and DOT1L proteins interact physically by co-immunoprecipitation (Supplementary Fig. 8e). These results imply that DOT1L and AR cooperate to regulate MYC expression.

Next, we performed ChIP-qPCR to assess the impact of DOT1L inhibition on enrichment of AR, DOT1L, and the other enhancer marks at the *MYC* enhancer (Fig. 7e). EPZ treatment of LNCaP cells led to a reduction in the recruitment of AR, DOT1L and Pol II as well as a reduction in H3K27ac, H3K4me2, and H3K79me2 marks at the *MYC* enhancer (Fig. 7e). Similarly, in C42B cells, EPZ treatment led to loss of AR and H3K79me2 at the enhancer (Supplementary Fig. 8f). These results indicate that DOT1L is required for AR enrichment at this distal *MYC* enhancer in order to regulate MYC expression in AR-positive cells.

Next, we examined if AR overexpression (Supplementary Fig. 8g) can rescue the loss of enhancer marks and MYC mRNA expression after DOT1L inhibition. In addition to increased enrichment of AR itself at the enhancer, we observed significantly higher enrichment of DOT1L, H3K79me2, H3K4me2, H3K27ac, and Pol II in the AR overexpressing cells when compared to the control cells (Fig. 7f). Moreover, AR overexpression was able to rescue the loss of binding of DOT1L and Pol II as well as the loss

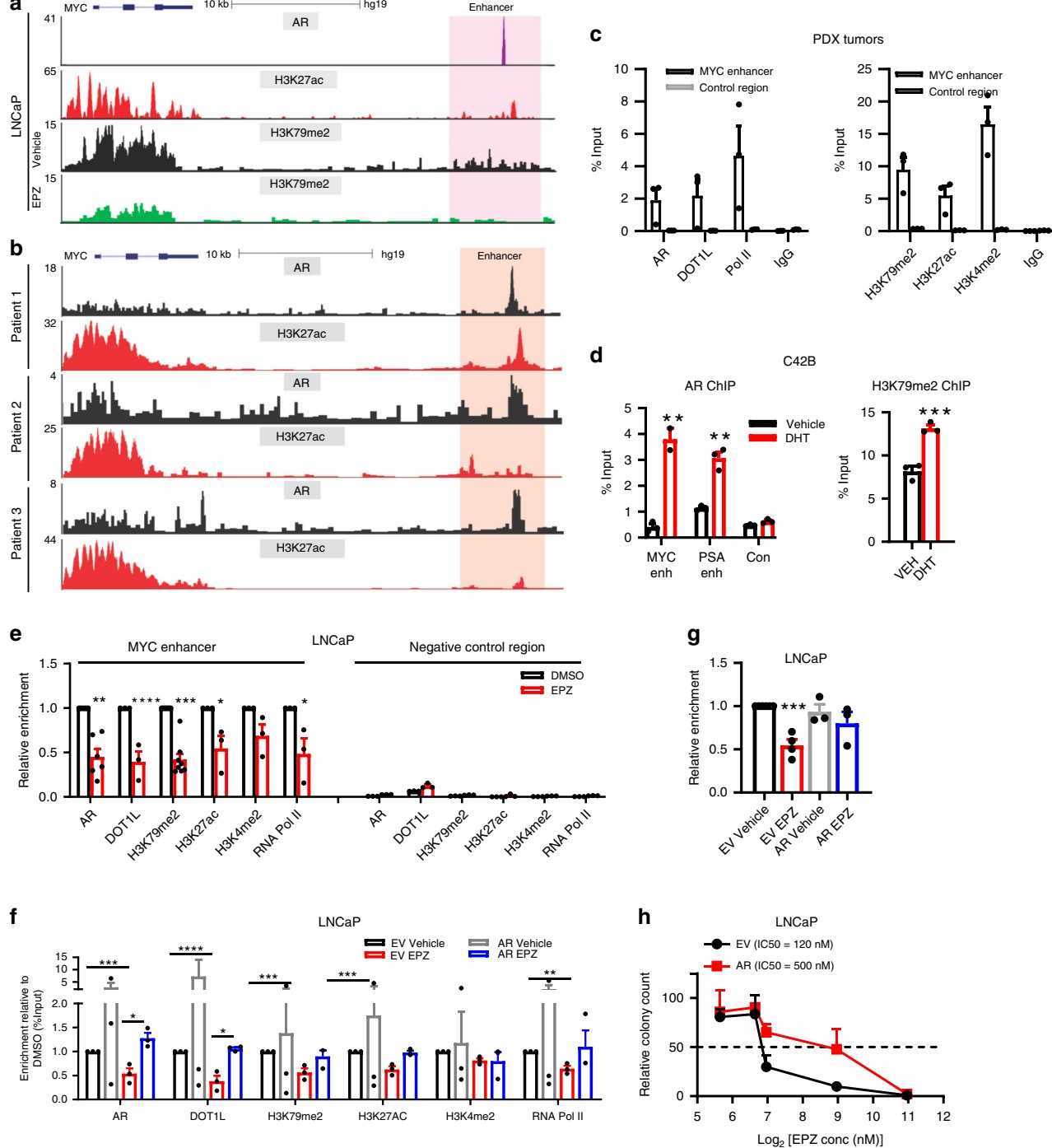

of enhancer marks caused by EPZ treatment (Fig. 7f). Importantly, the reduction in MYC mRNA caused by EPZ treatment was also by AR overexpression (Fig. 7g). Functionally, AR overexpressing LNCaP cells exhibited increased resistance to EPZ treatment (Fig. 7h). Finally, to show that this enhancer is critical for MYC expression, we used CRISPR-Cas9 to delete the AR-binding sequence in the *MYC* enhancer by using a guide RNA pair flanking the AR-binding site (Fig. 7i and Supplementary Fig. 8h). The viability of LNCaP cells was significantly decreased upon CRISPR-Cas9 deletion of the AR-binding site, while PC3 cells were unaffected (Fig. 7j). Furthermore, a concurrent loss of *MYC* expression was observed in the LNCaP cells but not the PC3 cells (Fig. 7k). Importantly, the expression of exogenous MYC in LNCaP cells rescued the loss of cell viability observed upon

deletion of the AR-binding site by CRISPR-Cas9 (Fig. 7i–k and Supplementary Fig. 8i). Thus, the downstream AR-binding region contributes to the regulation of MYC expression in LNCaP cells. Collectively, these results indicate that DOT1L and AR coregulation of *MYC* expression via a distal enhancer modulates sensitivity to DOT1L inhibition in PCa cells.

## Discussion

In this study, we have identified DOT1L as a promising therapeutic target whose inhibition selectively impairs the viability of AR-positive PCa. By targeting DOT1L using small molecule inhibitors and shRNA constructs in vitro and in vivo, we have shown that growth of AR expressing cells is selectively inhibited

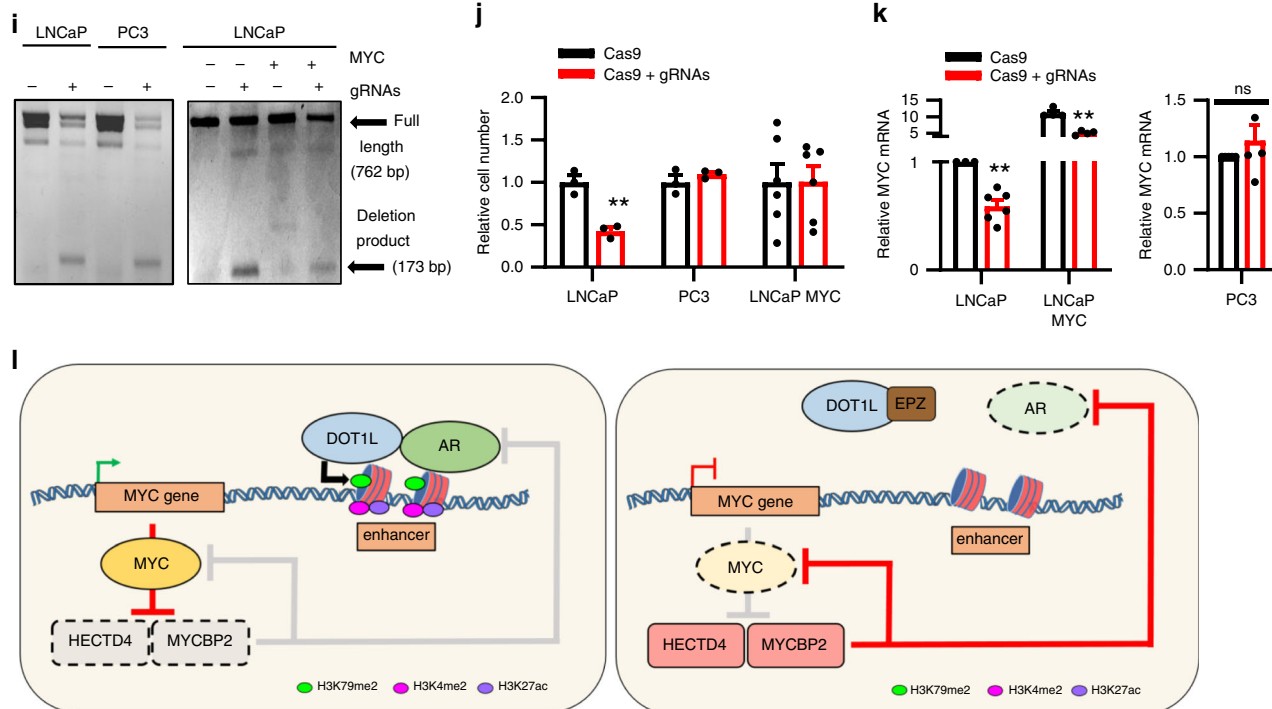

**Fig. 7 DOT1L and AR co-regulate MYC expression through a distant enhancer. a** ChIP-seq tracks of AR in LNCaP cells treated with R1881 (GSM353644), H3K27ac in LNCaP cells (GSM686937), H3K79me2 in Vehicle and EPZ-treated LNCaP cells (EPZ 1μM, 8 days), (top—bottom). **b** ChIP-seq tracks of AR and H3K27ac in 3 patient samples (GSE120738) at the *MYC* enhancer. **c** Enrichment of AR, DOT1L, H3K79me2, H3K27ac, H3K4me2 and RNA Pol II at *MYC* enhancer in PDX tumors. **d** AR and H3K79me2 enrichment at *MYC* enhancer in LNCaP cells treated with Vehicle or DHT for 3 h. **e** Enrichment of AR, DOT1L, H3K79me2, H3K27ac, H3K4me2 and RNA Pol II at the *MYC* enhancer in LNCaP cells treated with Vehicle or 1 μM EPZ for 8 days. **f** Enrichment of AR, DOT1L, H3K79me2, H3K27ac, H3K4me2 and RNA Pol II at the *MYC* enhancer in LNCaP cells transduced with EV or AR and treated with Vehicle or 1 μM EPZ for 8 days. **g** *MYC* expression in LNCaP EV or AR cells treated with Vehicle or EPZ for 8 days. **h** Quantitation of colony formation assays in LNCaP EV or AR cells treated with Vehicle or 1 μM EPZ for 12 days. **i** Results of genomic DNA PCR following CRISPR-Cas9-mediated deletion of AR-binding region in LNCaP, PC3, and LNCaP-MYC cells leading to a wild type full length band product and a deletion product. Representative image shown from 3 independent experiments. **j** Cells per FOV plotted for LNCaP, PC3 and LNCaP MYC (*n* = 5) transfected with gRNAs for CRISPR-Cas9-mediated deletion of the AR-binding site at the *MYC* enhancer after 5 days relative to Cas9 alone cells. **k** Relative *MYC* mRNA expression in LNCaP, PC3 (*n* = 5), and LNCaP-MYC (*n* = 5) transfected with or without gRNAs for CRISPR-Cas9 deletion of *MYC* enhancer after 3 days. **l** Graphical summary of DOT1L dependent regulation of *MYC* expression and its association with AR and MYC protein stability. Dashed lines represent decreased abundance. Gray lines represent inactive pathways. Statistical tests: P value determined by two-tailed *t* test (**c–h**, **j**, **k**). Error bars represent S.E.M. *n* = 3 biological replicates. *$p < 0.05$; **$p < 0.01$; ***$p < 0.001$; ****$p < 0.0001$.

when compared to AR-negative PCa cells. We found that H3K79 methylation is severely inhibited upon DOT1L inhibition or knockdown in both AR-positive and AR-negative cells. Our findings indicate that while some H3K79me2 enriched sites are shared between the two, a majority are unique and correlate with transcriptional programs specific to each cell type. This study also sheds light on the H3K79me2 landscape of AR-positive and AR-negative PCa cells.

We found that the AR and MYC pathways are inhibited subsequent to the loss of H3K79 methylation. Interestingly, the loss of AR protein led to a paradoxical increase in a subset of AR target genes indicating a complex transcriptional regulation network that is influenced by other transcription factors like MYC but ultimately dependent on DOT1L loss.

Both AR and MYC pathways were suppressed due to the loss of MYC/AR protein stability. We identified several E3 ligases as strong candidates responsible for AR and MYC protein loss as HECTD4, MYCBP2, and TRIM49. HECTD4 and MYCBP2 target AR and MYC for degradation while TRIM49 appears to promote AR and MYC stability. We have shown that these E3 ligases in turn are directly regulated by MYC.

The basis for the increased sensitivity of AR-positive PCa to DOT1L inhibition appears to be the coregulation of *MYC* by

DOT1L and AR at a distal enhancer active only in AR-positive PCa cells. Our results showed that DOT1L inhibition suppresses both AR and MYC pathways (Fig. 7l). We showed that AR and DOT1L are bound to this enhancer and regulate its function. We also found evidence that AR and DOT1L interact physically which further supports the model of their coregulation. A similar interaction has been observed between the estrogen receptor and DOT1L in breast cancer cells[11], suggesting a more general role for DOT1L in regulating nuclear receptors. While we do not exclude the possibility that DOT1L might methylate AR and affect its function, we note that reduced cell viability upon DOTL1 inhibition are only seen after H3K79me2 is lost. When DOT1L is inhibited, its displacement from the enhancer along with loss of AR and H3K79me2 at this site suppresses *MYC* expression. MYC in turn represses the expression of ubiquitin ligases, HECTD4 and MYCBP2 that promote AR and MYC protein degradation, further suppressing MYC and AR in a feed forward loop. We have also shown that increased AR expression can rescue the effects of DOT1L inhibition on cellular viability by restoring *MYC* expression. Furthermore, CRISPR-Cas9 deletion of this enhancer led to the loss of viability in LNCaP cells, which can be rescued by exogenous MYC expression indicating the critical function of this enhancer in regulating MYC expression.

A recent report identified a class of H3K79 methylated enhancers (KEE enhancers) indicating a role for H3K79 methylation in promoting promoter-enhancer contacts[46]. Our results show that the AR-bound distal *MYC* enhancer is a member of this subset of KEE enhancers that requires K79 methylation for maintaining chromatin accessibility, histone acetylation and transcription factor binding. Moreover, analyses of Pol II ChIA-PET maps has revealed that this downstream region of *MYC* interacts with the *MYC* promoter in AR-positive VCaP cells but not in AR-negative RWPE-1 or DU145 cells[47]. Another recent study also suggested a role for H3K79 methylation at the *MYC* promoter/gene in colorectal cancer cells[12]. However, while this study supports our observation that DOT1L plays a role in *MYC* gene regulation, our data showing lack of reduction of MYC or loss of viability despite loss of K79 methylation in PC3 cells contrasts with this model.

In summary, our studies indicate that DOT1L inhibition may be a viable therapeutic strategy for AR-positive PCa, including CRPC and ENZA resistant PCa.

## Methods

**Human prostate cancer data**. Gene expression data were downloaded from the NCBI Geo for the following datasets: Grasso PCa dataset [GSE35988; benign ($n = 28$), PCa ($n = 94$)][48], Yu PCa dataset [GSE6919; Normal ($n = 60$), PCa ($n = 81$)][49,50], Varambally PCa dataset [GSE3325; benign ($n = 6$), PCa ($n = 13$)][51], Taylor PCa dataset [GSE21032; Normal ($n = 29$), PCa ($n = 150$)][52], Roudier PCa dataset [GSE74367; primary cancers ($n = 11$), metastatic cancers ($n = 45$)][53], Gulzar PCa dataset [GSE40272; normal ($n = 66$), PCa ($n = 83$)][54] and from cbioportal.org: Beltran dataset [CRPC adeno ($n = 15$), CRPC-NE ($n = 34$)][55]. $p$ value determined by Welch's $t$ test. Gleason score data was used from TCGA dataset and Ross-Adams Discovery set. $p$ values were determined by ANOVA.

Survival analysis was performed using data from NCBI Geo and 25 percentile and 75 percentile were used as cutoffs from the Luca CancerMap PCa dataset [GSE94767; ($n = 46$ each)][56], Ross-Adams Discovery dataset [GSE70768; ($n = 27$ each)][57] and from cbioportal.org: TCGA Provisional PCa dataset [($n = 122$ each)][58], Grasso PCa dataset [GSE35988; cutoff at 75 percentile, high ($n = 11$), low ($n = 37$)][48], Ross-Adams (Validation) dataset [GSE70768; cutoff at 50th percentile, high ($n = 46$), low ($n = 46$)][57] and Gulzar PCa dataset [GSE40272; cutoff at 50th percentile, high ($n = 41$), low ($n = 41$)][54] and TCGA Gleason 7 patients [($n = 61$ each)][58]. Disease-free survival analysis for HECTD4 and MYCBP2 was done in MSKCC dataset[52] [cutoff at 10th percentile for HECTD4 and MYCBP2]. Disease-free survival analysis of TRIM49 was done in MSKCC dataset [Cutoff at median]. Patient survival curves were assessed by the Kaplan–Meier method. $p$ values were determined by log-rank test.

For Correlation analysis, the gene expression of DOT1L and MYC were analyzed using Spearman's rank correlation. Data used for this analysis were from the MSKCC dataset ($n = 150$)[52] downloaded from cbioportal.org. For E3 ligase correlation, data used from the TCGA Provisional dataset ($n = 499$)[58], SU2C dataset ($n = 118$)[59], MSKCC dataset[52] (n = 150), FHCCC dataset[60] ($n = 176$) downloaded from cbioportal.org. For correlation analysis between AR and MYC, samples with MYC or AR amplifications were excluded. Data was used from the TCGA dataset ($n = 336$)[58] and MSKCC dataset[52] ($n = 101$).

**Clinical samples**. RNA samples from normal prostate tissue ($n = 15$) and hormone dependent PCa ($n = 15$) were obtained from PCBN Repository. Thirty RNA samples from PCa metastases were kindly provided from Dr. Colm Morrissey from University of Washington, WA. All samples were de-identified and in compliance with ethical regulations and the approval of Institutional Review Board of University of Washington. All tissues were collected from patients who had signed written informed consent at the University of Washington.

**Immunohistochemistry**. A TMA with 80 cases and matched normal including a range of Gleason grade & pathology stage was acquired from PCBN. The TMA was processed for IHC staining as described. H3K79me2 antibody from Abcam was used. After primary antibody incubation, slides were incubated with ImmPRESS HRP anti-rabbit (Vector#MP-7401). Expression was visualized by using AEC peroxidase substrate (Vector #SK-4200). Slides were incubated with Hematoxylin (Vector #3404) and mounted with Glycergel Mounting Medium (Dako #C056330-2). Images were visualized in TissueGnostics microscope at Northwestern Core facility Center for Advanced Microscopy. The number of epithelial cells showing nuclear staining was estimated per core and scaled: 0, no positive cells; 1, 1–25% positive cells; 2, 26–50% positive cells; 3, 51–75% positive cells; and 4, 76–100% positive cells. These scores were multiplied with an intensity scale (1, weak; 2, moderate; and 3, intensive staining), and the mean staining for a patient was calculated. TMAs were scored by three investigators in a blinded fashion (RV, VS, and BL).

**Cell lines**. LNCaP, C42B, 22Rv1, PC3, and DU145 cells were grown in RPMI 1640 media (Gibco Life Technologies no. 11875-093) supplemented with 10% fetal bovine serum (FBS)—(Life Technologies no. 10437-028) and 1% penicillin/streptomycin antibiotic solution (Life Technologies no. 15140-122). RWPE-1 cells were grown in keratinocyte serum-free media supplemented with 0.05 mg/ml bovine pituitary extract, 5 ng/ml epidermal growth factor (Thermo Fischer Scientific no. 17005042) and 1% penicillin/streptomycin antibiotic solution. For assays in charcoal stripped medium, cells were first hormone starved in RPMI 1640 media without Phenol red supplemented with 10% charcoal stripped FBS and 1% penicillin/streptomycin antibiotic solution for 48 h before start of assay. All cells were verified as mycoplasma free and genetically authenticated by ATCC.

**Mice**. NOD-SCID mice (Jackson Laboratory) used in this study were housed in a pathogen-free animal barrier facility or a containment facility, as appropriate. When mice were 6–8 weeks old, they were injected subcutaneously with 2 million live LNCaP cells pretreated with either DMSO control or 10 μM EPZ for 7 days (100 μl, 1:1 with matrigel). Tumor volume was measured using calipers until they reached 1500 mm³. For the PDX experiment, prostate PDX model (TM00298) and NSG mice were obtained from Jackson Laboratory. All experiments and procedures were performed in compliance with ethical regulations and the approval of the Northwestern University Institutional Animal Care and Use Committee.

**PDX organoid assays**. Patient-derived xenografts (PDX) model of PCa (TM00298) was obtained from The Jackson Laboratory. To generate PDX organoids[61], tumors were minced and digested with collagenase (Gibco) in RPMI 1640 media with 10% FBS for 2 h at 37 °C and incubated with Trypsin for 5 min at 37 °C. Subsequently, digested tissues were treated with DNase I (Sigma), and then passed through 40 μm cell strainers to obtain single cells. Cells were resuspended in organoid culture media composed of Hepatocyte medium (Corning) supplemented with 10 ng/ml epidermal growth factor (Corning), 5% heat-inactivated charcoal-stripped FBS, 1× Glutamax (Gibco), 5% matrigel (Corning), 10 μM ROCK inhibitor (Y-27632, STEMCELL Technologies), 10 nM DHT (Sigma) and primocin (Invivogen). Cells were plated in Ultra-Low Attachment Surface plates (Corning) at 5000 cells per 100 μl media containing either DMSO, EPZ4777 (1 μM, 10 μM) or EPZ5676 (1 μM). For lentiviral transduction, dissociated PDX cells were transduced with shControl or shDOT1L on day 0. On day 1, virus containing media was replaced with organoid media and cells were replated in Ultra-Low Attachment Surface plates. Additional 100 μl media was added at day 4 and day 8. On day 8, organoids were centrifuged at 300 rcf for 5 min and collected for protein analysis. On day 12, 3D viability was assessed using CellTiter-Glo® 3D Cell Viability Assay (Promega). Cell line organoid culture was performed similar to PDX organoids[61–63]. Briefly, 2000 cells were resuspended in organoid media containing low percentage matrigel (5%) then plated in to 96-well ultralow attachment plates (Corning no. 3474).

**Cell growth assays**. For long-term clonogenic assays, cells were counted and plated in low density in six-well plates and treated with different concentrations of EPZ004777 (Epizyme) or EPZ5676 (Selleckchem). After 12 days, colonies were fixed and stained with crystal violet and photographed. To assess cell viability, cells were plated in 10 cm culture dishes and treated with DMSO control or EPZ for 10 days. Media was replenished every 3–4 days. On day 10, cells were trypsinized, counted and replated in 96-well plates (5000 cells per well). On day 12, viability was assessed using the CellTiter 96 AQueous One Solution Cell Proliferation Assay (Promega) as per manufacturer's instructions.

**DNA constructs, lentivirus production, and lentiviral transduction of cell lines**. MSCB-hDot1Lwt was a gift from Dr. Yi Zhang (Addgene plasmid # 74173). pcDNA3.1 plasmid (EV) and Flag tagged AR plasmid was kindly provided by Dr. Meejeon Roh. pSMP-Luc and pSMP-Dot1L_1 was a gift from George Daley (Addgene plasmid # 36339 and plasmid # 36394). FM1-YFP and FM1-AR-YFP were used as described previously[61]. Halotag-HECTD4 was obtained from Promega (FHC24891). TRIM49 construct was obtained from GenScript (OHu03301D). pCMV-Pam-FL was a gift from Vijaya Ramesh (Addgene plasmid # 42570). MSCV_Cas9_puro was a gift from Christopher Vakoc (Addgene plasmid # 65655). pCS2-MYC plasmid was made by Dr. Huiying Han[30].

Viral particles were produced in 293T cells transfected with the expressing vector, Δ8.9 packaging vector (for lentivirus) or MMLV packaging vector (for retrovirus) and VSVG envelope vector (2:1:1) using Lipofectamine 2000 (Invitrogen) in Opti-MEM media (Gibco) as described[64]. LNCaP cells were transduced with the virus and 1 μg/ml of puromycin was added to select stably expressing cells.

**qRT-PCR**. Total RNA extraction was performed using the TRIzol reagent (Life Technologies, Rockville, MD), according to the manufacturer's instructions. The reverse transcriptase polymerase chain reaction (RT-PCR) was performed using Moloney murine leukemia virus reverse transcriptase (Invitrogen) as per manufacturer's instructions. qRT-PCR was performed using PowerUp SYBR Green Master Mix (Applied Biosystems). Primers used for qRT-PCR are listed in Supplementary Table 1.

**Microarray.** LNCaP and PC3 cells were treated with DMSO control or 1 μM EPZ for 8 days. Total RNA extraction was performed using the TRIzol reagent (Life Technologies), according to the manufacturer's instructions. RNA was cleaned up using the RNeasy Plus Mini Kit (Qiagen). RNA was submitted to the Genomics Core at the University of Illinois, Chicago for microarray analysis. RNA was hybridized to GeneChip® Human Transcriptome 2.0 arrays and raw data (CEL files) were processed and normalized using Robust Multichip Average by Bioconductor oligo package. Differentially expressed genes were identified by the Bioconductor limma package. Heatmap view of differentially expressed genes was created by Cluster and Java Treeview. GSEA was done using C2 curated and C6 oncogenic gene sets.

**RNA interference.** For HECTD4 knockdown, two siRNAs were tested from Dharmacon (1# D-018270-01 and 2# D-018270-02). HECTD4 siRNA #1 was used for subsequent studies. For MYCBP2 knockdown, three siRNAs were tested (2# D-006951-01 and 3# D-006951-02, 1# Qiagen SI00109235). MYCBP2 siRNA #2 was used for subsequent studies. Cells were transiently transfected with HECTD4 (D-018270-01), MYCBP2 (D-006951-01), TRIM49 (D-007030-01), HERC3 (D-007179-01) siRNA or Non-targeted siRNA (Dharmacon Catalog no. D-001210-01-05) and DharmaFECT transfection reagent (Dharmacon), according to the manufacturer's protocol. Cells were analyzed for all experiments after 48 h. For DOT1L knockdown, cells were transiently transfected with DOT1L targeting siRNA (Dharmacon MU-014900-01-0002) or Non-targeted siRNA (Dharmacon Catalog no. D-001210-01-05) and DharmaFECT transfection reagent (Dharmacon), according to the manufacturer's protocol. MYC knockdown (Dharmacon D-003282-15-0005) was done as per manufacturer's instructions[63].

**ChIP.** LNCaP and PC3 cells were treated with DMSO control or 1 μM EPZ for 8 days. Cells were washed with PBS and crosslinked with 1% formaldehyde for 10 mins. PDX tumors were harvested, rinsed in PBS, and crosslinked at room temperature for 30 min in 1% formaldehyde. ChIP was then performed as before[65]. For ChIP-qPCR, primers used are listed in Supplementary Table 2. For ChIP-seq, barcoded sequencing libraries were generated using KAPA Library Preparation Kits (Kapa Biosystems) according to the manufacturer's protocol. Libraries were quantified using a qPCR-based quantification (Kapa Biosystems) and sequenced on a NextSeq 500 instrument (Illumina). Sequence reads were aligned to the Human Reference Genome (assembly hg19) using Burrows–Wheeler Alignment Tool Version 0.6.1. Peak identification, overlapping, subtraction and feature annotation of enriched regions were performed using Hypergeometric Optimization of Motif EnRichment suite. Weighted venn diagrams were created by R package Vennerable. Differentially enriched genes were loaded into Enrichr website for ChEA.

**Flow cytometry.** LNCaP cells were transfected with ARE-GFP construct (Gentarget LVP912-R) using Lipofectamine and treated with vehicle or EPZ for 8 days. On day 8, single-cell dissociation of LNCaP cells and flow cytometry was performed as described previously[64]. The gating strategy is shown in Supplementary Fig. 9.

**Western blot.** Western blotting was carried out as described[66,67]. Cell lysates were prepared in RIPA buffer (Sigma) with protease and phosphatase inhibitors. Lysates were quantified and run by SDS-PAGE and transferred to PVDF membranes. Membranes were then blocked and exposed to the following antibodies: H3K79me2 (ab177184; Abcam,1:1000), AR (RB-9030-P1; Thermo Fischer, 1:1000), PSA (A0562; Dako, 1:1000), Actin (sc-1616; Santa Cruz Biotech, 1:1000), Histone H3 (ab1791; Abcam, 1:3000), GAPDH (sc-20357; Santa Cruz Biotech, 1:1000), DOT1L (EMD Millipore MABE425, Abcam ab72454, 1:100), MYC (Abcam ab32072, 1:1000), Halotag (Promega G9211, 1:1000), Flag (Sigma-Aldrich F1804, 1:1000), Myc-Tag (Cell Signaling Technology 2276S, 1:1000), c-Myc (phospho S62) (Abcam ab51156, 1:1000) and Anti-c-Myc (phospho T58, Abcam ab185655, 1:1000) and Ubiquitin (Cell Signaling Technology 3936, 1:1000). Blots were then imaged using chemiluminescent substrate (Thermo Scientific) and ChemiDoc Imaging System (Bio-Rad).

**Co-immunoprecipitation assays.** Briefly, cells were lysed using NETN buffer (20 mM Tris HCl [pH 8.0], 100 mM NaCl, 1 mM EDTA, and 0.5% Nonidet P-40). The insoluble pellets from the crude lysis step was treated with Enzymatic shearing cocktail from Nuclear Complex Co-IP Kit(Active Motif) for 90 min at 4 °C to release nuclear proteins. Both cell lysis fractions were combined and lysates were incubated with either AR (RB-9030-P1; Thermo Fischer) or Flag antibody (F1804; Sigma-Aldrich) overnight. Magnetic beads were added to the cell lysate and incubated for 2 h. Beads were washed and bound fractions were eluted with 2× loading buffer. The eluted fractions were then analyzed by Western blotting.

**CRISPR-Cas9 editing.** To stably express CAS9 in LNCaP and PC3 cells, we generated a CAS9 (Streptococcus pyogenes CRISPR-Cas) expressing lentivirus (Addgene 65655) from 293T cells. Lentiviral infection efficacy was >90% and cells were maintained with 2 μg/ml puromycin. Two synthetic guide RNAs (gRNAs) (CRISPR crRNA, Integrated DNA Technologies) were designed using the CRISPR Design Tool (crispr.mit.edu), those with off-target effects were excluded. gRNAs were ordered from Integrated DNA technologies as g-blocks (455 bp fragment) that contains U6 promoter, gRNA target sequence, guide RNA scaffold and a termination signal. The sequence for gRNA 1 and 2 is CCCCCTGGTTGTCAAACT CTGGG and GCCTCCCATCAGTCATCCCAG-GG, respectively. They were delivered by transient transfection reagent Lipofectamine. Enhancer knockout was confirmed by genomic DNA PCR using the following primer sequences: F primer —TAAAGGAAAAGGGACTGTGGAA, R primer—CAGGTCTTCTCAGGTCTT TGCT.

**Mass spectrometric analysis.** LC–MS/MS analysis was performed by Northwestern Proteomics Core Facility. Briefly, LNCaP cells were treated with Vehicle or 1 μM EPZ for 8 days followed by treatment with MG-132 for 6 h. Cells were then lysed as described above. Lysates were quantified and equal amounts of protein were incubated with AR (RB-9030-P1; Thermo Fischer) overnight. Magnetic beads were added to the cell lysate and incubated for 2 h. Beads were washed and bound fractions were eluted with 2× loading buffer. The pulldown samples were loaded onto stacking gel for 5 min, and gel lane holding the total loaded proteins was cut and submitted to the facility. The proteins were digested with trypsin and analyzed by LC–MS/MS using a Dionex UltiMate 3000 Rapid Separation nanoLC and an Orbitrap Elite Mass Spectrometer (Thermo Fisher Scientific Inc, San Jose, CA) following the standard protocol in the Proteomics Core Facility. Scaffold (version Scaffold_4.8.6, Proteome Software Inc., Portland, OR) was used to validate MS/MS based peptide and protein identifications. Peptide identifications were accepted if they could be established at >90.0% probability by the Peptide Prophet algorithm with Scaffold delta-mass correction. Protein identifications were accepted if they could be established at >99.0% probability and contained at least two identified peptides. Protein probabilities were assigned by the Protein Prophet algorithm. Proteins that contained similar peptides and could not be differentiated based on MS/MS analysis alone were grouped to satisfy the principles of parsimony. Protein probabilities were assigned by the Protein Prophet. Proteins that contained similar peptides and could not be differentiated based on MS/MS analysis alone were grouped to satisfy the principles of parsimony (version Scaffold_4.8.9). All identified proteins were filtered by Contaminant Repository for Affinity Purification (CRAPome) database following the workflow 1 instructions (www.crapome.org). The proteins with over 20% frequency in CRAPome database were considered as nonspecific bindings and removed from the list.

**Statistical analyses.** Statistical analyses were performed using unpaired two-tailed Student's $t$ test. Survival studies were analyzed by Log-rank (Mantel–Cox) test. Correlations were analyzed by Spearman's correlation coefficient ($r$). Data are presented as mean ± standard error of the mean (S.E.M.), unless otherwise indicated. For all analyses, results were considered statistically significant with $p < 0.05$, $*p < 0.05$, $**p < 0.01$, $***p < 0.001$, and $****p < 0.0001$.

**Reporting summary.** Further information on research design is available in the Nature Research Reporting Summary linked to this article.

## Data availability
The data that support the findings of this study (Microarray and ChIP-seq) are available in the GEO under accession GSE135575. ChIP-seq data is deposited in GSE135574 and Microarray data is deposited in GSE135573. The expression and ChIP-seq data referenced in this study is available from the NCBI GEO website. The ChIP-seq tracks of AR in LNCaP cells treated with R1881 (GSM353644) was downloaded from GSE14092. H3K27ac in LNCaP cells (GSM686937) is from GSE27823. ChIP-seq tracks of AR and H3K27ac in three patient samples (223T, 227T, and 229T) was extracted from GSE120738. MYC ChIP-seq track was downloaded from GSE73994. Gene expression data were downloaded from the NCBI Geo from GSE35988, GSE6919, GSE3325, GSE21032, GSE94767 GSE70768, GSE40272, GSE74367, GSE3971, GSE20842, GSE7696, GSE9348, GSE13159, and GSE13507. Source data underlying all figures provided as Source data file. All the other data supporting the findings of this study are available within the article and its Supplementary Information files and from the corresponding author upon reasonable request.

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

## Acknowledgements

We thank Epizyme for EPZ004777, Dr. Colm Morrissey for clinical RNA samples, PCBN for clinical RNA samples and TMA, Northwestern Core facility Center for Advanced Microscopy for TMA analysis, Robert H. Lurie Comprehensive Cancer Center Flow Cytometry Core for help in flow cytometry analysis. This work was supported by grants RO1CA123484, RO1CA196270, and P50CA180995 from the NCI; and by the Department of Defense Prostate Cancer Research Program Awards W81XWH-14-2-0182, W81XWH-14-2-0183, W81XWH-14-2-0185, W81XWH-14-2-0186, and W81XWH-15-2-0062 to the Prostate Cancer Biorepository Network (PCBN). R.V. was supported by the Chicago Cancer Baseball Charites Award (T32 CA080621) and J.C.Z. was supported by NCI grant R50CA211271.

## Author contributions

R.V. and S.A.A. conceived and designed the experiments. R.V. designed and performed most of the in vitro and in vivo biological experiments. K.U. performed PDX organoid assays. V.S performed the TMA IHC and cell line organoid assays. Y.R. performed colony formation assays and flow cytometry. J.C.Z. conducted the ChIP-seq and microarray analysis. K.U., S.P., B.L, J.A., H.H., Y.A.Y, M.T., and Z.R.C. assisted in biological studies and data analysis. D.C., F.G., J.Y., and B.C. contributed to data interpretation. S.A.A. supervised the overall project. R.V. and S.A.A. wrote the paper, with input from all the other authors.

## Competing interests

The authors declare no competing interests.
