## [Peer Review File · Nature Communications]

Reviewers' comments:

Reviewer #1 (Remarks to the Author):

In this manuscript, Vatapalli, et al., investigated the potential efficacy of targeting the lysine methyltransferase DOT1L, which is known to catalyze the methylation of lysine 79 of histone 3 (H3K79me1, me2 and me3), in androgen receptor-positive prostate cancer. They report an upregulation of DOT1L expression in tumor samples compared to normal samples and a correlation between high DOT1L expression and poorer patient outcome (survival) (Fig 1.). They next demonstrated that treatment with a small molecule inhibitor of DOT1L leads to a specific inhibition of proliferation in AR-dependent prostate cancers (Fig 2). The authors performed ChIP-seq analyses to examine H3K79me2 occupancy genome-wide, and observed the expected global decrease in H3K79me3 across all transcriptional start sites (Fig. 2K/L). The authors then describe an apparent effect of DOT1L inhibition on both AR (Fig. 3) and MYC (Fig. 4) protein levels. The authors try to explain these effects by investigating the expression of different E3 ubiquitin ligases following DOT1L inhibitor treatment (Fig. 5) and focus specifically on HECTD4, MYCBP2 and TRIM49. Finally, the authors try to equate the regulation and function of these E3 enzymes back to AR, MYC and DOT1L function in prostate cancer and wrap up the manuscript by describing an apparent role for AR and DOT1L in activating a downstream “enhancer” of the MYC gene.

Although a role for DOT1L in prostate cancer could be interesting and of medical relevance, I'm afraid I cannot recommend this manuscript for further consideration for publication in Nature Communications. Overall, the manuscript does not provide any further mechanistic insight into DOT1L or AR function and lacks both focus and scientific rigor. Some specific critiques include:

1. The authors fail to recognize and cite seminal work directly relevant to this study from a number of laboratories around the world (see PMID: 30171029, 30472189, 30616689, 30775443). Notably, their findings are apparently at odds with other published data. While this is perfectly fine, at least some discussion of these differences must be provided.
2. In Fig. 1b the authors use different cut-offs (90th, 75th, 50th percentiles) for the various datasets. How do they justify this?
3. In Fig. 1C it is unclear what the units are (“mRNA expression”).
4. Zoomed pictures are needed to examine the quality and reliability of IHC data provided in Fig. 1D. Here also, the apparent heterogeneity observed in the boxplot (Fig. 1D, left panel) should be addressed.
5. Throughout the manuscript the authors are very inconsistent with the concentrations of DOT1L inhibitor utilized. Consistent with their studies showing decreased proliferation of LnCAP cells already at 250 nM, the on-target IC50 values for EPZ5676 in cells lies well below 1 uM. Given the specificity of the inhibitor, even concentrations of 1 uM should still provide relatively specific, on-target effects, but findings such as those in Fig. 2H and Fig. 3E (more problematic...) suggest that much of what the investigators report may be due to off-target effects of the inhibitor, which can only be observed at extremely high concentrations.
6. In Fig. 2K and L, the authors demonstrate a global decrease in H3K79me2 levels at the TSS of genes genome-wide. Given the fact that DOT1L is the only methyltransferase for H3K79, this finding is both expected (and an essential positive control). Particularly problematic are the authors' interpretation of “changes” in H3K79me2 levels at genes. It is well established that H3K79me2 (or H3K79me3) levels are directly related to gene expression levels. Thus, any gene specifically expressed in one cell type will display higher levels of H3K79me2 in its gene body compared to another cell type where that gene is not expressed. The same could be said for any epigenetic mark (H3K27ac, H3K36me3, H2B monoubiquitylation) or core transcriptional complex

member (e.g., RNA Polymerase II). Thus, data displaying specific marking of genes simply reflects the transcriptional program(s) at work within a given cell type. Thus, the marking of androgen-responsive genes in LNCAP vs. PC3 cells is rather a positive control than anything else. Moreover, since new H3K79 methylation is completely blocked by the concentration of EPZ provided, any residual methylation on genes must be interpreted with extreme caution. Since there is no known H3K79 demethylase, as pointed out by the Crabtree group, the loss of H3K79me2 from chromatin has to do with nucleosome turnover. Based on this, no much of anything can be concluded from the Venn diagrams in Fig. 2 M. At best, a better (i.e., quantitative, based on intensity) analysis could provide some information on degree of H3K79me2 loss over time, but does not provide any significant mechanistic insight.

7. It is unclear what the difference is between the Venn diagrams shown in Fig. 2M vs. Supp. Fig. 2H. It appears that one (Fig. 2M) may be “peaks” across the genome (irrespective of relationship to genes), whereas the other (Supp. 2H) may be “genes”. This needs to be clearly described and the differences/similarities need to be discussed. In particular, it appears that most “marked genes” (Supp. 2H) are lost after EPZ treatment (...the expected effect). This would then beg the question whether gene-associated H3K79me2-enriched regions lose the mark quicker than intergenic regions (which may be expected if they have a higher rate of nucleosome turnover).

8. Related to 6 and 5, above, nowhere in the manuscript do the authors provide any demonstration of the mechanistic importance of H3K79me2 to the observed changes. In fact, their WBs seem to be at odds with this. While they seek to connect DOT1L methyltransferase activity, and thereby H3K79me2, with gene (and protein) expression changes, the authors actually see different effects of DOT1L inhibitor treatment in different experiments. In Fig. 2H they see some inhibition of H3K79me2 at 1 uM, in Fig. 2J the more expected strong effect of the same inhibitor concentration, but in Fig. 3E no effect of the inhibitor on PSA expression. In fact, this finding appears to be somewhat at odds with the results in 3F.

9. Many of the described effects are very minimal. This is particularly apparent in Fig. 3H, where the decrease in AR-dependent genes such as PSA, TMPRSS2 and KLK2 is very minimal (again...at odds with what they authors describe in other portions of the manuscript).

10. The authors try to claim a specific function of DOT1L regulation of MYC in AR-positive tumors and state that MYC-related signatures were only seen in LNCAP cells. However, their table in Supp. Fig. 4 clearly demonstrates a “MYC targets repressed by serum” signature as their third hit.

11. Particularly problematic are the analyses of both AR (Fig. 3D) and MYC (Fig. 4G) half-lives. The graphic representation (right panels in the respective figures) do not even closely reflect what is obvious in the blots. In both cases, the protein levels in the vehicle treated cells are much higher than in the treated samples (i.e., lower starting material). Both graphic representations lead the reader to believe that the protein levels stay stable after cycloheximide treatment. This clearly is NOT the case. For example, the AR levels appear to be half (or less than) already at 2 h after cycloheximide treatment in the vehicle condition and probably 10-fold or more lower after 24 h. The graph suggests that no nearly no change is observed, with the levels at 24 h being well above 50% of the starting levels. While the differences are not quite as extreme in Fig. 4G, the interpretation still appears very problematic. The MYC levels under vehicle condition appear to be half (or less!) already after 0.5 h, which is shown as no change (or increase???) in the graph on the right. Thus, I find these data and the erroneous conclusions drawn from them, to completely lack robustness and scientific rigor.

12. In figure 5 the authors then seek to identify E3 ubiquitin ligases controlling AR and MYC stability. Here, no mechanistic explanation is provided at all as to why these ligases should be increased at the mRNA level. Thus, the only thing shown is a correlation (i.e., the mRNA levels go up somehow), without having any direct connection to DOT1L (presumably they also display H3K79me2 in the control condition, which go down after treatment...but why do the mRNA levels

go up?). This appears to be merely correlative without any explanation

13. Also, it is odd that, while the authors see effects with HECTD4 or MYCBP2 knockdown in Fig. 5D, they only utilize combined knockdowns in 5E and 5F, but leave out MYCBP2 altogether in 5G and H.

14. Knockdowns must also be performed with at least two different individual siRNAs for each targeted gene to ensure specificity of the effects.

15. The results in Fig. 5H also suggest that global polyubiquitination increases after HECTD4 overexpression (WB: Ub in Input samples). How can this be? The global ubiquitination should be regulated by a large number of different ubiquitin ligases so that the overexpression of a single one would be expected to have little or no effect on the global level of ubiquitination, but a big effect on specific ubiquitination targets. This needs to at least be addressed. Also, Flag-AR expression is missing as a positive control as well.

16. In Fig. 5G the appropriate controls are missing. The authors only show co-IPs under conditions where the proteins (Flag-AR and Halotag-HECTD4) are co-expressed. All combinations should be shown so that it can, for example, be clearly demonstrated that Flag-AR only comes down with the Halo-tag when it is co-expressed with Halotag-HECTD4 and vice versa.

17. Also, the rationale (and underlying mechanisms) for including TRIM49 are completely unclear to me. The authors should be more clearly focused. How the expression of a ubiquitin ligase induces protein stability (or the other way around – knockdown inducing instability) is completely unclear and would rely upon indirect mechanisms not discussed by the authors. While potentially interesting, this appears completely unrelated to the rest of the manuscript.

18. The value of the data in Fig. 6 are unclear. While the authors try to connect MYC regulation to AR, the apparent activation (Fig. 6A) or inhibition (6B) has no effect on HECTD4 expression and fairly moderate effects on MYCBP2 mRNA levels (...protein levels should be shown as well, as should AR and MYC levels). Then, in Fig. 6C and D, it appears the effects of MYC knockdown on HECTD4 and MYCBP2 are universal and independent of AR. This leaves the question remaining whether the described effects are specific or universal??

19. I find Fig. 7 to also be particularly problematic. The authors describe an “enhancer” 20 kb downstream of the MYC gene. While MYC expression is particularly well established to be dependent upon the activity of distal enhancers, the authors do not really provide any evidence that this region is, in fact, an enhancer. Actual ChIP-seq data (publically available) should be provided for H3K27ac (not just peaks) as well as H3K4me1 and H3K4me3. Furthermore:

a. The essential controls are missing from Fig. 7B (IgG, H3K79me2 as well as primers detecting a non-occupied site).

b. The effects in Fig. 7C are VERY mild.

c. In Fig. 7D, the only apparent change that is robust is the decrease in MYC expression (also VERY mild) after enzalutamide treatment. The authors' claim that “DOT1L overexpression in LNCaP cells rescued the loss of MYC expression seen upon ENZA treatment” is completely unfounded (differences not significant and error bars clearly overlapping).

d. Fig. 7E is also very problematic. The authors label the Z-axis as “Enrichment relative to DMSO (% input)”. Are the “relative” or are they “% input”. It appears that they are, indeed, relative values with the vehicle-treated samples set to 1. Thus, it is impossible to conclude whether there is any significant enrichment at these regions. This must be shown as (real) percent input, with the appropriate controls (see above, IgG as a negative control, as well as primers for a region not enriched with the proteins). It is also unclear how the authors mechanistically envisage that catalytic inhibition of DOT1L (...over a period of 8 days...thereby likely inducing many secondary or tertiary effects) actually leads to all the described changes. Is this a direct effect of DOT1L at these

sites? Or a result of decreased AR protein levels? This is completely unclear and not addressed by the authors.

e. In Fig. 7F, apart from the effects of EPZ, the remaining effects are very moderate and the combination of the small effect and the high error preclude the authors or reader from making any conclusions. A claim that AR overexpression rescues the effects is an extreme overstatement.

f. Fig. 7G has all the same problems described above for Fig. 7E. In addition, I find it extremely difficult to interpret much from this figure. It is clear that AR and DOT1L occupancy increases with AR overexpression, but even this is very dramatically reduced by EPZ treatment (although above the EV / Vehicle condition, the degree of decrease is still much higher than in the EV / Vehicle vs. EV / EPZ comparison). Moreover, the large error bars preclude drawing any real conclusions from the data.

g. In Fig. 7H the authors appear to have mislabeled the X axis. It is shown as "Log2 [EPZ concentration]". By convention this would be molar concentrations, whereas the authors appear to have used nanomolar concentrations. Also here the effects appear very mild. Additional results would be essential to provide convincing data.

20. The legend of Supp. Fig. 6 lists "d" and "e". However, the version of the figure provided only has panels A-C. Also, panel "C" is incorrectly described in the legend.

Reviewer #2 (Remarks to the Author):

In this paper, the authors make the very interesting observation that DOT1L is overexpressed in poor prognosis prostate cancer as well as several other solid tumours and this is associated with increased H3K79me2 levels. They go on to show that only tumours that are androgen receptor (AR) positive are actually sensitive to DOT1L inhibitors in vitro and this is also replicated by one in vivo xenograft experiment. They then explore this mechanism and find that DOT1L inhibition decreases AR levels and activity, and that the effect is ultimately controlled by AR mediated activation of MYC through an H3K79me2 dependent enhancer, and via MYC regulation of several E3 ligases. Overall, these experiments are very interesting and it is quite impressive how the authors managed to weave their way through several observation to get to the final conclusion. There are however some points that need to be clarified.

Major points

1. For most of the paper, two cell lines were analyzed. In particular, for most of the paper from Figure 2k onwards, LNCaP cells play the role of representing all AR positive samples. How much of this data can be replicated in other AR+ (and AR-) cell lines and in primary patient material? I am not necessarily suggesting that the genomic approaches in Figure 2k-I need to be performed in multiple cell lines and in primary material, just that it would be useful to know how generalizable at least some of the mechanistic observations in the paper are, particularly the dependence of MYC expression on AR and DOT1L in AR positive tumours.

2. Are MYC levels different in AR positive vs AR negative cells? The reason for asking this is that it would be useful to know if AR dependence causes MYC overexpression. The alternative is that MYC is also actively expressed in AR negative cells, but the gene is activated through an AR independent mechanism. Figure 4d would seem to predict that MYC should be slightly overexpressed in AR+ samples, although potentially still active in AR- samples.

3. There is an additional MYC enhancer almost 2Mb away from the promoter (Shi et al, Genes Dev. 2013 27(24):2648-62. doi: 10.1101/gad.232710.113 and Bahr et al, Nature. 2018 553(7689):515-520. doi: 10.1038/nature25193). In LNCaP and PC3 cells is this enhancer marked by H3K79me2 and does it contribute to regulation of MYC in AR positive cells? Does DOT1L inhibitor treatment cause a loss of H3K27ac in this region and are there additional AR binding sites in the region of the 2Mb enhancer in LNCaP cells? AR ChIP-seq in LNCaP cells would help determine if there are other AR binding sites in additional MYC enhancers in the region.

Minor points

1. In Figure 2m and S2h in the comparison between EPZ treated LNCaP and PC3, are the Venn diagrams just showing H3K79me2 peaks that change upon DOT1L inhibition, or are these peaks that are left over after inhibition? Some sample CHIP-seq tracks to illustrate the trends would be helpful here.
2. The westerns in Fig 3a are overexposed and not completely convincing. A lower exposure so that most of the bands are in the linear range and some quantification of the signal would be helpful.
3. Do AR positive vs negative show different levels of H3K79me2, or are these the same across the samples?

Reviewer #3 (Remarks to the Author):

The manuscript by Vatapalli et al. identifies a novel DOT1L-AR-MYC signaling axis that governs AR-dependent prostate cancer cell growth and that is targetable with DOT1L inhibitors. Their data suggests that AR-negative cells are not dependent on this pathway. The authors also provide new data that identifies specific E3 ubiquitin ligases that are implicated in DOT1L-dependent AR and MYC protein stability. The authors have generated a significant amount of interesting and compelling data however, there are a few concerns/questions that are detailed below.

1. In Figure 2m and Supp Fig 2h, what accounts for the new H3K79me2 peaks in the EPZ treated cells? What is the significance of these new peaks and do they overlap between LNCaP and PC3 cells?
2. The authors determine that there is no impact of DOT1L on AR gene regulation but are they also ruling out a physical interaction between the 2 proteins? This is particularly relevant considering the data in Fig 7 and co-localization of DOT1L and AR at the MYC enhancer.
3. It is not clear why it takes 8 days to observe the drop in AR (and MYC) protein? The cycloheximide data in Fig 3d and 4g, respectively suggests that these proteins turnover by 24 hours. A time course of AR and MYC expression +/- EPZ would help.
4. Based on Fig 3b growth curve it is suggested that DOT1L promotes androgen independent growth in LNCaP cells. It would be informative if these experiments can be done in castrate resistant, AR-positive cells (e.g. 22Rv1, C42B).
5. What is the status of MYC, HECTD4 and MYCBP2 in DOT1L over-expressing cells (Fig 3b)?
6. The data in Figs 4 and 5 suggest that DOT1L dependent MYC loss is dependent on AR activity – but this is only shown in one cell line (LNCaP) – what about castrate-resistant cells (e.g. C42B or 22Rv1)? What is the level of increase in AR and MYC protein in Fig 5d following siHECTD4/MYCBP2?
7. The mass spectrometry data showed 4 peptides corresponding to MYCBP2 interacting with AR. This needs co-IP validation. In addition, can the authors comment why FOXA1 or HOXB13 or other known AR interacting proteins are not on the list?
8. What is the clinical relevance of HECTD4, MYCBP2 and TRIM49 expression? How do the levels of expression for these genes correlate with DOT1L and MYC in clinical samples?
9. Is the known MYC protein degradation SCFFbxw7 ubiquitin ligase or the MYC T58, S62 residues implicated in the context of DOT1L-MYC axis?
10. Does MYC bind to the HECTD4, MYCBP2 promoters?

Minor issues:

1. Higher resolution IHC data is needed in Fig 1d.
2. Show protein levels for data shown in Sup Fig 2g.

3. In the legend for Fig 4a, "Barfield study" should be "Barfeld study".
4. The authors need to quantitate the level of DOT1L protein reduction in Fig 4f following shDOT1L? The western blot shown is not terribly convincing.
5. Statistical significance needs to be assessed for data in Fig 6a,b.
6. What is the level of MYC knock-down in Fig 6c,d?

Below we provide a point-by-point response to specific reviewer comments:

Reviewer #1

1. The authors fail to recognize and cite seminal work directly relevant to this study from a number of laboratories around the world (see PMID: 30171029, 30472189, 30616689, 30775443). Notably, their findings are apparently

at odds with other published data. While this is perfectly fine, at least some discussion of these differences must be provided.

- We thank the reviewer for pointing out these studies and have cited these papers accordingly. The findings do not necessarily contradict our thesis, since we are not claiming that AR positivity is the only factor that sensitizes tumor to DOT1L inhibition.

2. In Fig. 1b the authors use different cut-offs (90th, 75th, 50th percentiles) for the various datasets. How do they justify this?

- These studies were all done independent of one another and were performed on different platforms, which is why one could not directly compare expression levels across studies without rigorous normalization. The cut-offs merely reflected levels that separated high from low DOT1L expressing samples within each study.

3. In Fig. 1C it is unclear what the units are (“mRNA expression”).

- This is now corrected to “mRNA expression (relative to b-actin)”.

4. Zoomed pictures are needed to examine the quality and reliability of IHC data provided in Fig. 1D. Here also, the apparent heterogeneity observed in the boxplot (Fig. 1D, left panel) should be addressed.

- High-power images are now included. Heterogeneity in intensity is due to variability in K79 methylation levels across tumors, which is to be expected.

5. Throughout the manuscript the authors are very inconsistent with the concentrations of DOT1L inhibitor utilized. Consistent with their studies showing decreased proliferation of LnCAP cells already at 250 nM, the on-target IC50 values for EPZ5676 in cells lies well below 1 uM. Given the specificity of the inhibitor, even concentrations of 1 uM should still provide relatively specific, on-target effects, but findings such as those in Fig. 2H and Fig. 3E (more problematic...) suggest that much of what the investigators report may be due to off-target effects of the inhibitor, which can only be observed at extremely high concentrations.

- We have used RNAi to validate our findings as shown **Fig 2e, 2f, 2j, 3c, 3l, 4h** ruling out off-target effects of the DOT1L inhibitor. We have also performed the converse overexpression studies (e.g. **Fig 3e, 4i**). Regarding the findings in previous figure 2h and 3e, these experiments were done at the earlier time point of 8d in contrast to the viability assays which were done at 12d. In **Supplementary Fig 2d,e,f, 3a, 4c**, we show that the effects on AR and MYC levels and ultimately cell viability are time-dependent, consistent with the fact that loss of K79 methylation is by nucleosome turnover. The reviewer pointed out that K79me2 is lost by nucleosome turnover in comment #6 below.

6. In Fig. 2K and L, the authors demonstrate a global decrease in H3K79me2 levels at the TSS of genes genome-wide. Given the fact that DOT1L is the only methyltransferase for H3K79, this finding is both expected (and an essential positive control). Particularly problematic are the authors' interpretation of “changes” in H3K79me2 levels at genes. It is well established that H3K79me2 (or H3K79me3) levels are directly related to gene expression levels. Thus, any gene specifically expressed in one cell type will display higher levels of H3K79me2 in its gene body compared to another cell type where that gene is not expressed. The same could be said for any epigenetic mark (H3K27ac, H3K36me3, H2B monoubiquitylation) or core transcriptional complex member (e.g., RNA Polymerase II). Thus, data displaying specific marking of genes simply reflects the transcriptional program(s) at work within a given cell type. Thus, the marking of androgen-responsive genes in LNCAP vs. PC3 cells is rather a

positive control than anything else. Moreover, since new H3K79 methylation is completely blocked by the concentration of EPZ provided, any residual methylation on genes must be interpreted with extreme caution. Since there is no known H3K79 demethylase, as pointed out by the Crabtree group, the loss of H3K79me2 from chromatin has to do with nucleosome turnover. Based on this, no much of anything can be concluded from the Venn diagrams in Fig. 2 M. At best, a better (i.e., quantitative, based on intensity) analysis could provide some information on degree of H3K79me2 loss over time, but does not provide any significant mechanistic insight.

- We also interpret the data as the reviewer did. However, the main point of performing this experiment is to show that in PC3 cells which are resistant to DOT1L inhibitor, the inhibitor is able to get into cells and modulate H3K79 methylation. This rules out trivial explanations for the lack of effect of the inhibitor, e.g. lack of inhibitor uptake or enhanced efflux in PC3. So this entire experiment was meant as an important control experiment to show that the inhibitor could inhibit global K79 methylation as effectively in PC3 cells as it does in LNCaP cells. Of course, the particular K79 methylation sites affected will be different in the two cell lines based on their distinct transcriptional programs.

7. It is unclear what the difference is between the Venn diagrams shown in Fig. 2M vs. Supp. Fig. 2H. It appears that one (Fig. 2M) may be “peaks” across the genome (irrespective of relationship to genes), whereas the other (Supp. 2H) may be “genes”. This needs to be clearly described and the differences/similarities need to be discussed. In particular, it appears that most “marked genes” (Supp. 2H) are lost after EPZ treatment (...the expected effect). This would then beg the question whether gene-associated H3K79me2-enriched regions lose the mark quicker than intergenic regions (which may be expected if they have a higher rate of nucleosome turnover).

- Yes **Fig. 2m** represent “peaks” while previous figure 2h, now **Supplementary Fig 2i** represents “genes” as the reviewer indicated. We apologize for not making this clearer. The difference in number is due to presence of peaks in intergenic regions and multiple peaks called within the gene. We have looked at whether gene lose K97 methylation more completely than intergenic regions as the reviewer alluded to. We find, that while in LNCaP both intergenic and genic regions lose the H3K79 methylation at the same rate, in PC3 the genic regions lose the H3K79me2 mark at a higher rate than intergenic regions. (**Supplementary Figure 2k**).

8. Related to 6 and 5, above, nowhere in the manuscript do the authors provide any demonstration of the mechanistic importance of H3K79me2 to the observed changes. In fact, their WBs seem to be at odds with this. While they seek to connect DOT1L methyltransferase activity, and thereby H3K79me2, with gene (and protein) expression changes, the authors actually see different effects of DOT1L inhibitor treatment in different experiments. In Fig.2H they see some inhibition of H3K79me2 at 1 uM, in Fig. 2J the more expected strong effect of the same inhibitor concentration, but in Fig. 3E no effect of the inhibitor on PSA expression. In fact, this finding appears to be somewhat at odds with the results in 3F.

- The inhibitors used in previous figures 2h and 2j (now **Fig 2i and 2j**) are distinct DOT1L inhibitors with different potencies, which explains the difference in loss of H3K79 methylation at 1uM. The western blot for PSA in previous Fig 3e which was overexposed has been updated (**Fig. 3h**) and showing a clear effect at 1uM. Furthermore, the strong effect seen in **Fig. 3i** (previously Fig 3f) is due to the high sensitivity of the reporter assay.

9. Many of the described effects are very minimal. This is particularly apparent in Fig. 3H, where the decrease in AR-dependent genes such as PSA, TMPRSS2 and KLK2 is very minimal (again...at odds with what they authors describe in other portions of the manuscript).

- As discussed in the Introduction section of this letter, analysis of early time DOT1L inhibition or AR inhibition (without DOT1L inhibition) will be expected to only partially modulate the pathway, thus the modest effects. In other words, loss of K79 methylation is critical for changes in MYC expression and in turn the expression of the ubiquitin ligases that degrade AR and MYC. The data that show more robust effects were from longer term inhibitor treatment after the feed-forward loop has been established.

10. The authors try to claim a specific function of DOT1L regulation of MYC in AR-positive tumors and state that MYC-related signatures were only seen in LNCaP cells. However, their table in Supp. Fig. 4 clearly demonstrates a “MYC targets repressed by serum” signature as their third hit.

- The “MYC targets repressed by serum signature” that appeared in the GSEA analysis of PC3 cells were not validated as indicating an effect on MYC activity in these cells for the following reasons: **1)** Other canonical MYC signatures were not modulated in PC3 cells, unlike in LNCaP cells. **2)** The leading edge genes from this “MYC targets repressed by serum signature” were not exclusively MYC target genes. In fact our analysis showed that these genes are also equally regulated by modulation of other factors such as NR2C and SOX17 (**Supplementary Fig 3b**). **3)** Crucially, MYC expression was **not** decreased in PC3 cells after DOT1L inhibition (**Fig 4e and Supplementary Fig 4g**). **4)** Cell viability was also not reduced after DOT1L inhibition, which would have been the expected consequence of reduced MYC activity. We therefore conclude that these “MYC targets repressed by serum” signature genes were likely modulated due to effects on other regulators in PC3 cells.

11. Particularly problematic are the analyses of both AR (Fig. 3D) and MYC (Fig. 4G) half-lives. The graphic representation (right panels in the respective figures) do not even closely reflect what is obvious in the blots. In both cases, the protein levels in the vehicle treated cells are much higher than in the treated samples (i.e., lower starting material). Both graphic representations lead the reader to believe that the protein levels stay stable after cycloheximide treatment. This clearly is NOT the case. For example, the AR levels appear to be half (or less than) already at 2 h after cycloheximide treatment in the vehicle condition and probably 10-fold or more lower after 24 h. The graph suggests that no nearly no change is observed, with the levels at 24 h being well above 50% of the starting levels. While the differences are not quite as extreme in Fig. 4G, the interpretation still appears very problematic. The MYC levels under vehicle condition appear to be half (or less!) already after 0.5 h, which is shown as no change (or increase???) in the graph on the right. Thus, I find these data and the erroneous conclusions drawn from them, to completely lack robustness and scientific rigor.

- The fact that modulation of the ubiquitin ligases that modulate AR and MYC protein levels is seen **after** K79 methylation is lost meant that the protein half-life studies had to be done in cells treated with DOT1L inhibition for at least 8 days, at which point AR and MYC protein levels are decreased. The CHX chase is done to assess the stability of the **existing** AR and MYC protein pools in the vehicle versus DOT1L inhibitor treated cells. Protein levels for each condition at the different time points were normalized to the corresponding time zero for that treatment. For the AR stability blots in **Fig. 3g**, we have performed additional replicates and updated the figure, showing a half-life of approximately 3 hours in EPZ477-treated cells compared to about 9.4 hours for vehicle treated cells. The half-life of AR in LNCaP cells is consistent with the literature (1, 2).

12. In figure 5 the authors then seek to identify E3 ubiquitin ligases controlling AR and MYC stability. Here, no mechanistic explanation is provided at all as to why these ligases should be increased at the mRNA level. Thus, the only thing shown is a correlation (i.e., the mRNA levels go up somehow), without having any direct connection to DOT1L (presumably they also display H3K79me2 in the control condition, which go down after treatment...but why do the mRNA levels go up?). This appears to be merely correlative without any explanation.

- Our data supports a model wherein DOT1L inhibition targets MYC only in AR-positive cells (and thus the downstream E3 ligases) because of the downstream AR-bound enhancer. The E3 ubiquitin ligases are repressed by MYC (**Fig. 6a-b**). We now show that that MYC binds to the regulatory regions of these genes by ChIP (**Fig. 6c-d**). Direct knockdown of MYC in PC3 cells does modulate the expression of the E3 ligases as expected. However in EPZ treated PC3 cells, there is no loss of MYC to begin with and hence the expression of these ubiquitin ligases are not altered.

13. Also, it is odd that, while the authors see effects with HECTD4 or MYCBP2 knockdown in Fig. 5D, they only utilize combined knockdowns in 5E and 5F, but leave out MYCBP2 altogether in 5G and H.

- In previous Fig 5d (new **Fig. 5c**), we tested all the candidate E3 ligases for evidence of effect of AR/MYC protein levels. In subsequent studies we combined the two validated, upregulated E3 ligases HECTD4 and MYCBP2 to recapitulate the effect that more than one E3 ligase is altered by DOT1L inhibition. For protein-protein interaction studies in **Fig. 5f-h** we now show interactions with AR since MYCBP2's interaction with MYC has been well characterized.

14. Knockdowns must also be performed with at least two different individual siRNAs for each targeted gene to ensure specificity of the effects.

- To address RNAi specificity, we have tested additional siRNAs have similar effects ensuring specificity (**Supplementary Figure 5d**), and have also performed the converse overexpression studies (**Supplementary Figure 5h**).

15. The results in Fig. 5H also suggest that global polyubiquitination increases after HECTD4 overexpression (WB: Ub in Input samples). How can this be? The global ubiquitination should be regulated by a large number of different ubiquitin ligases so that the overexpression of a single one would be expected to have little or no effect on the global level of ubiquitination, but a big effect on specific ubiquitination targets. This needs to at least be addressed. Also, Flag-AR expression is missing as a positive control as well.

- Since HECTD4 is expected to affect the ubiquitination of many other targets in addition to AR, it is not surprising that a global increase in polyubiquitination is observed. Flag-AR lane has been added to the new figure (now **Figure 5g**).

16. In Fig. 5G the appropriate controls are missing. The authors only show co-IPs under conditions where the proteins (Flag-AR and Halotag-HECTD4) are co-expressed. All combinations should be shown so that it can, for example, be clearly demonstrated that Flag-AR only comes down with the Halo-tag when it is co-expressed with Halotag-HECTD4 and vice versa.

- These controls have now been included in the figure as requested (now **Figure 5f**).

17. Also, the rationale (and underlying mechanisms) for including TRIM49 are completely unclear to me. The authors should be more clearly focused. How the expression of a ubiquitin ligase induces protein stability (or the other way around – knockdown inducing instability) is completely unclear and would rely upon indirect

mechanisms not discussed by the authors. While potentially interesting, this appears completely unrelated to the rest of the manuscript.

- Our analysis that searched for putative E3 ligases that are dysregulated by DOT1L inhibition identified HECTD4 and MYCBP2 to be upregulated and TRIM49 to be downregulated. Some E3 ligases can stabilize proteins. For example, TRIM39 has been shown to stabilize the proteins MOAP-1 and Cactin by modulating polyubiquitination (3, 4). Thus our data showing that knockdown of TRIM49 decreases AR and MYC stability are in line with these studies.

18. The value of the data in Fig. 6 are unclear. While the authors try to connect MYC regulation to AR, the apparent activation (Fig. 6A) or inhibition (6B) has no effect on HECTD4 expression and fairly moderate effects on MYCBP2 mRNA levels (...protein levels should be shown as well, as should AR and MYC levels). Then, in Fig. 6C and D, it appears the effects of MYC knockdown on HECTD4 and MYCBP2 are universal and independent of AR. This leaves the question remaining whether the described effects are specific or universal??

- MYC regulates HECTD4 and MYCBP2 directly as shown by ChIP and gene expression data. This occurs in both AR-positive and AR-negative cells, so from this perspective it is universal. We are not claiming that this regulation only happens in AR-positive cells. However, DOT1L and AR coregulate MYC in AR-positive cells, which is why DOT1L inhibition leads to changes in MYC expression and protein stability and the MYC targets HECTD4 and MYCBP2 only in AR-positive cells. DOT1L inhibition does not regulate MYC in AR-negative cells because they lack the AR-binding enhancer, so there are no changes in MYC or its targets MYCBP2 and HECTD4. However, knocking down MYC itself in PC3 cells affects HECTD4 and MYCBP2 expression. Therefore, while the effects of MYC on the ubiquitin ligases is universal, the effects of DOT1L inhibition via a DOT1L/AR/MYC pathway is only seen in AR positive cells and is AR specific.

19. I find Fig. 7 to also be particularly problematic. The authors describe an “enhancer” 20 kb downstream of the MYC gene. While MYC expression is particularly well established to be dependent upon the activity of distal enhancers, the authors do not really provide any evidence that this region is, in fact, an enhancer. Actual ChIP-seq data (publically available) should be provided for H3K27ac (not just peaks) as well as H3K4me1 and H3K4me3.

- A previous study has documented AR binding to this region and implicated it in MYC regulation as we pointed out (5). We have added new figures with showing binding of AR and H3K27ac at this region in human prostate cancer samples (**Fig. 7b**) and PDX tumors (**Fig. 7c**).
- We have also used CRISPR/Cas9 to delete the ARE in this region in LNCaP and PC3 cells. Notably, deletion of this region led to significant cell death of LNCaP cells while the viability of PC3 cells was unaffected. Notably, the viability of LNCaP cells with deletion in this ARE can be rescued by exogenous MYC expression (**Fig. 7i,j. Supplementary Fig 8h**). This provides functional support for the role of this enhancer in regulating MYC expression in AR-positive cells.

Furthermore:

a. The essential controls are missing from Fig. 7B (IgG, H3K79me2 as well as primers detecting a non-occupied site).

- All the controls have now been updated (new **Fig 7c,d,e**).

b. The effects in Fig. 7C are VERY mild.

- This figure (now 7g) shows effect of AR inhibition by enzalutamide. As discussed earlier, since K79 methylation was not inhibited this effect is in fact to be expected.

c. In Fig. 7D, the only apparent change that is robust is the decrease in MYC expression (also VERY mild) after enzalutamide treatment. The authors' claim that "DOT1L overexpression in LNCaP cells rescued the loss of MYC expression seen upon ENZA treatment" is completely unfounded (differences not significant and error bars clearly overlapping).

- While the decrease in MYC expression after enzalutamide treatment is mild, it is statistically significant. This is consistent with our model that both K79me2 and AR binding are important for regulating MYC expression. Inhibiting AR without inhibiting K79me2 is predicted to have modest effects.

d. Fig. 7E is also very problematic. The authors label the Z-axis as "Enrichment relative to DMSO (% input)". Are the "relative" or are they "% input". It appears that they are, indeed, relative values with the vehicle-treated samples set to 1. Thus, it is impossible to conclude whether there is any significant enrichment at these regions. This must be shown as (real) percent input, with the appropriate controls (see above, IgG as a negative control, as well as primers for a region not enriched with the proteins). It is also unclear how the authors mechanistically envisage that catalytic inhibition of DOT1L (...over a period of 8 days...thereby likely inducing many secondary or tertiary effects) actually leads to all the described changes. Is this a direct effect of DOT1L at these sites? Or a result of decreased AR protein levels? This is completely unclear and not addressed by the authors.

- The Y-axis label for this figure had a typo: "% input". This is now corrected. These are relative values as the reviewer surmised. We have added the additional controls. Mechanistically, it is established when the binding of a transcriptional activator is inhibited, activation marks on chromatin can be reduced and repression marks may even appear. In this case, loss of AR protein as well as loss of K79 methylation, which is known to mark some enhancers as recently reported (6) are the likely reasons for the observed effects.

e. In Fig. 7F, apart from the effects of EPZ, the remaining effects are very moderate and the combination of the small effect and the high error preclude the authors or reader from making any conclusions. A claim that AR overexpression rescues the effects is an extreme overstatement.

- As the reviewer noted, the effects of EPZ are pronounced but this was only observed in control but not in the corresponding AR-overexpressing cells. This is consistent with increased levels of AR sustaining MYC expression in the face of EPZ treatment.

f. Fig. 7G has all the same problems described above for Fig. 7E. In addition, I find it extremely difficult to interpret much from this figure. It is clear that AR and DOT1L occupancy increases with AR overexpression, but even this is very dramatically reduced by EPZ treatment (although above the EV / Vehicle condition, the degree of decrease is still much higher than in the EV / Vehicle vs. EV / EPZ comparison). Moreover, the large error bars preclude drawing any real conclusions from the data.

- The data show that EPZ-treated AR-overexpressing cells retained higher levels of AR, DOT1L and the associated chromatin marks compared to EPZ-treated control cells. EPZ did reduce factor binding and chromatin marks in AR-overexpressing cells relative to vehicle since AR degradation is still occurring in these cells (the AR expressed is not degradation resistant). However, since these cells have higher AR levels, there is still significant binding of AR, DOT1L, and K79me etc. even after EPZ-treatment which translates to increased resistance to EPZ (but not total resistance). This addresses the findings in Fig. 7h in the critique in "g" below.

g. In Fig. 7H the authors appear to have mislabeled the X axis. It is shown as “Log2 [EPZ concentration]”. By convention this would be molar concentrations, whereas the authors appear to have used nanomolar concentrations. Also here the effects appear very mild. Additional results would be essential to provide convincing data.

- We have corrected the X axis in this figure. Thank you for pointing it out. The magnitude of the effects is explained in the comment to “h” above.

20. The legend of Supp. Fig. 6 lists “d” and “e”. However, the version of the figure provided only has panels A-C. Also, panel “C” is incorrectly described in the legend.

- Thank you. This has now been corrected.

Reviewer #2

Major points

1. For most of the paper, two cell lines were analyzed. In particular, for most of the paper from Figure 2k onwards, LNCaP cells play the role of representing all AR positive samples. How much of this data can be replicated in other AR+ (and AR-) cell lines and in primary patient material? I am not necessarily suggesting that the genomic approaches in Figure 2k-l need to be performed in multiple cell lines and in primary material, just that it would be useful to know how generalizable at least some of the mechanistic observations in the paper are, particularly the dependence of MYC expression on AR and DOT1L in AR positive tumours.

- We have now added new data on 22Rv1 and C42B cell lines as well as patient-derived PDX tumor and organoid demonstrating that MYC expression is coregulated by DOT1L and AR. We have shown that DOT1L inhibition and shDOT1L leads to downregulation in AR and MYC expression in these models. We now also show that DOT1L upregulation leads to an MYC overexpression. (**Figure 2f,i, 3b,c,d, 4f,h,i, 7d, Supplementary Fig 4d,5c,7f**)
- We also show AR and enhancer marks at the MYC enhancer in PDX tumors and human prostate cancer samples (**Figure 7b-c**).

2. Are MYC levels different in AR positive vs AR negative cells? The reason for asking this is that it would be useful to know if AR dependence causes MYC overexpression. The alternative is that MYC is also actively expressed in AR negative cells, but the gene is activated through an AR independent mechanism. Figure 4d would seem to predict that MYC should be slightly overexpressed in AR+ samples, although potentially still active in AR- samples.

- This is an interesting point. We have examined human prostate cancer datasets and found that, after excluding samples with MYC copy number alterations, levels of MYC positively did correlate to AR mRNA levels (**Supplementary Fig 8a**). Similar observations have been made by other groups (5, 7). These data will therefore suggest that MYC expression is AR dependent in prostate cancer in the absence of MYC amplifications.

3. There is an additional MYC enhancer almost 2Mb away from the promoter (Shi et al, Genes Dev. 2013 27(24):2648-62. doi: 10.1101/gad.232710.113 and Bahr et al, Nature. 2018 553(7689):515-520. doi: 10.1038/nature25193). In LNCaP and PC3 cells is this enhancer marked by H3K79me2 and does it contribute to regulation of MYC in AR positive cells? Does DOT1L inhibitor treatment cause a loss of H3K27ac in this region and are there additional AR binding sites in the region of the 2Mb enhancer in LNCaP cells? AR ChIP-seq in LNCaP cells would help determine if there are other AR binding sites in additional MYC enhancers in the region.

- Thank you for pointing this out. We have examined this enhancer as the reviewer suggested and have found no evidence of H3K79me2, suggesting that it does not contribute to DOT1L-mediated regulation of MYC (**Figure Supplementary 7b-c**). Examination of ChIP-seq data from LNCaP and ChIP-qPCR analysis did not show AR or H3K27ac peaks at this particular site that could participate in MYC regulation. This can be explained by the difference in cell contexts since the previous studies were performed in mouse myeloma cell line RN2.

Minor points

1. In Figure 2m and S2h in the comparison between EPZ treated LNCaP and PC3, are the Venn diagrams just showing H3K79me2 peaks that change upon DOT1L inhibition, or are these peaks that are left over after inhibition? Some sample ChIP-seq tracks to illustrate the trends would be helpful here. These data indicate peaks that are left over after inhibition. Since, H3K79 methylation is replaced mainly by turnover, these new peaks might be attributed to positional 'scrambling' of histones containing 'old' K79 methylation during cell replication cycles.
2. The westerns in Fig 3a are overexposed and not completely convincing. A lower exposure so that most of the bands are in the linear range and some quantification of the signal would be helpful. Quantitation of the AR protein levels has now been added in **Fig 3a** along with AR western blot in additional cell lines showing that AR protein levels are reduced after DOT1L inhibition or knockdown.
3. Do AR positive vs negative show different levels of H3K79me2, or are these the same across the samples? There is no significant difference in levels of H3K79me2 between AR positive and AR negative cell lines (**Supplementary Fig. 2h**). We also saw no significant difference in the number of H3K79me2 marked sites in LNCaP and PC3 cells.

Reviewer #3

1. In Figure 2m and Supp Fig 2h, what accounts for the new H3K79me2 peaks in the EPZ treated cells? What is the significance of these new peaks and do they overlap between LNCaP and PC3 cells?
 - We were quite intrigued by this observation of new H3K79me2 peaks that emerged upon EPZ treatment. This might be attributed to positional 'scrambling' of histones containing 'old' K79 methylation during cell replication cycles (8). It is hypothesized that H3K79 methylation is lost by replacement with new unmethylated histone and the old histones with H3K79 are displaced from their original positions (9).
2. The authors determine that there is no impact of DOT1L on AR gene regulation but are they also ruling out a physical interaction between the 2 proteins? This is particularly relevant considering the data in Fig 7 and co-localization of DOT1L and AR at the MYC enhancer.
 - We found evidence that AR and DOT1L interact in co-IP studies in 293T cells (**Supplementary Figure 8e**).
3. It is not clear why it takes 8 days to observe the drop in AR (and MYC) protein? The cycloheximide data in Fig 3d and 4g, respectively suggests that these proteins turnover by 24 hours. A time course of AR and MYC expression +/- EPZ would help.

- This is mainly due to the fact that H3K79me2 at the *MYC* enhancer is lost slowly after inhibiting the DOT1L methyltransferase. It is established that H3K79 methylation is lost slowly after Dot1L inhibition over time through nucleosome turnover. Due to this, *MYC* expression is significantly reduced only after **8 days** of EPZ treatment (**Supplementary Figure 4c**). Our model shows that when DOT1L is inhibited, loss of H3K79 methylation at the enhancer leads to suppression of *MYC* expression. This reduction leads to an increase in the downstream ubiquitin ligases which target *MYC* and AR protein levels. Therefore, after EPZ treatment, decreased AR protein levels are observed at later time points.

4. Based on Fig 3b growth curve it is suggested that DOT1L promotes androgen independent growth in LNCaP cells. It would be informative if these experiments can be done in castrate resistant, AR-positive cells (e.g. 22Rv1, C42B).

- We have now done these experiments and we did not find a significant difference in the growth curves of C42B-EV and C42B-DOT1L expressing cells. This may be because these cells are already androgen independent and so DOT1L expression may not add an additional growth advantage in these conditions. However, in androgen dependent LNCaP cells DOT1L expression leads to AR overexpression giving rise to androgen independence.

5. What is the status of *MYC*, *HECTD4* and *MYCBP2* in DOT1L over-expressing cells (Fig 3b)?

- We have added these data in **Fig 4i** and **Supplementary Fig 5c**. The results show that DOT1L expression leads to an upregulation of *MYC* and downregulation of *HECTD4* and *MYCBP2* in LNCaP, C42B and 22rv1 cells. Hence, we can conclude that DOT1L positively regulates *MYC* levels and negatively regulates *HECTD4* and *MYCBP2* indirectly.

6. The data in Figs 4 and 5 suggest that DOT1L dependent *MYC* loss is dependent on AR activity – but this is only shown in one cell line (LNCaP) – what about castrate-resistant cells (e.g. C42B or 22Rv1)? What is the level of increase in AR and *MYC* protein in Fig 5d following siHECTD4/*MYCBP2*?

- We have now shown that DOT1L inhibition by EPZ leads to downregulation of *MYC* in castrate resistant cells like C42B and 22rv1 along with a PDX organoid model. (**Fig 4f,g,h** and **Supplementary Fig 4d**). This confirms that DOT1L dependent *MYC* loss is indeed dependent on the AR status of the cells.
- In the old figure 5d which is now **Fig 5c**. the increase in AR protein is 2 fold following siHECTD4 and siMYCBP2 while *MYC* is increased by 1.5 fold.

7. The mass spectrometry data showed 4 peptides corresponding to *MYCBP2* interacting with AR. This needs co-IP validation. In addition, can the authors comment why *FOXA1* or *HOXB13* or other known AR interacting proteins are not on the list?

- We have validated AR/*MYCBP2* interaction by co-IP (**Fig. 5h**). In our study, we have identified proteins like *PRKDC* that are known AR interacting proteins. *FOXA1* and *HOXB13* are not on the list, possibly as less abundant transcription factors. We are aware of the limitations of this assay and hence the interaction of *HECTD4* and *MYCBP2* with AR was assessed through pulldown assays and not through Mass spectrometry results alone (**Fig. 5f-h**).

8. What is the clinical relevance of HECTD4, MYCBP2 and TRIM49 expression? How do the levels of expression for these genes correlate with DOT1L and MYC in clinical samples?

- We have found that low MYCBP2 expression (expected to cause less degradation of AR and MYC) does correlate with poor survival in multiple datasets (**Supplementary Figure 5k**). We also found that high TRIM49 expression (which would stabilize AR and MYC) correlated with worse disease free survival in the Taylor /MSKCC dataset (**Supplementary Figure 5l**). We did not find significant association of HECTD4 expression with survival. We suspect that this might be due to the effects of other unknown HECTD4 targets that might influence survival of prostate cancer cells. We did not detect significant correlation between the Ubiquitin ligases with DOT1L perhaps due to the indirect nature of the regulation.

9. Is the known MYC protein degradation SCFFbxw7 ubiquitin ligase or the MYC T58, S62 residues implicated in the context of DOT1L-MYC axis?

- We thank the reviewer for this insightful comment. We have examined this axis by analyzing T58 and S62 phosphorylation after DOT1L inhibition. We did not find evidence of a change in these modifications (**Supplementary Fig. 4f**), suggesting that the degradation pathway is independent of SCFFbxw7 ligase complex but through that MYCBP2 and HECTD4 ligases.

10. Does MYC bind to the HECTD4, MYCBP2 promoters?

- We have assessed this by ChIP and found robust binding of MYC to the HECTD4 and MYCBP2 promoters (**Fig. 6c-d**). Thus MYC represses expression of HECTD4 and MYCBP2 by directly binding to their promoters.

Minor issues:

1. Higher resolution IHC data is needed in Fig 1d. This is added. 2. Show protein levels for data shown in Sup Fig 2g. We have added protein data for H3K79me showing equivalent levels in AR+ and AR- cells. 3. In the legend for Fig 4a, "Barfield study" should be "Barfeld study". This is corrected. 4. The authors need to quantitate the level of DOT1L protein reduction in Fig 4f following shDOT1L? The western blot shown is not terribly convincing. We have updated these data. We performed western blot for shDOT1L in LNCaP and additional cell lines that convincingly shows that MYC protein levels are dramatically reduced upon DOT1L knockdown. (**Figure 4h**). 5. Statistical significance needs to be assessed for data in Fig 6 a,b. This is done (**Supplementary Figure 6 a-b**). 6. What is the level of MYC knock-down in Fig 6c,d? This is now shown (**Figure 6b**).

We hope that you will agree that we have satisfactorily responded to all the concerns and that the manuscript is now acceptable for publication in Nature Communication.

Sincerely,

Sarki Abdulkadir

References:

1. Gregory CW, Johnson RT, Mohler JL, French FS, Wilson EM. Androgen receptor stabilization in recurrent prostate cancer is associated with hypersensitivity to low androgen. *Cancer Research*. 2001;61(7):2892-8.
2. Yeap BB, Krueger RG, Leedman PJ. Differential posttranscriptional regulation of androgen receptor gene expression by androgen in prostate and breast cancer cells. *Endocrinology*. 1999;140(7):3282-91.
3. Lee SS, Fu NY, Sukumaran SK, Wan KF, Wan Q, Yu VC. TRIM39 is a MOAP-1-binding protein that stabilizes MOAP-1 through inhibition of its poly-ubiquitination process. *Exp Cell Res*. 2009;315(7):1313-25.
4. Suzuki M, Watanabe M, Nakamaru Y, Takagi D, Takahashi H, Fukuda S, et al. TRIM39 negatively regulates the NFkappaB-mediated signaling pathway through stabilization of Cactin. *Cell Mol Life Sci*. 2016;73(5):1085-101.
5. Gao LN, Schwartzman J, Gibbs A, Lisac R, Kleinschmidt R, Wilmot B, et al. Androgen Receptor Promotes Ligand-Independent Prostate Cancer Progression through c-Myc Upregulation. *Plos One*. 2013;8(5).
6. Godfrey L, Crump NT, Thorne R, Lau IJ, Repapi E, Dimou D, et al. DOT1L inhibition reveals a distinct subset of enhancers dependent on H3K79 methylation. *Nature Communications*. 2019;10.
7. Bai S, Cao S, Jin L, Kobelski M, Schouest B, Wang X, et al. A positive role of c-Myc in regulating androgen receptor and its splice variants in prostate cancer. *Oncogene*. 2019;38(25):4977-89.
8. Sweet SM, Li M, Thomas PM, Durbin KR, Kelleher NL. Kinetics of re-establishing H3K79 methylation marks in global human chromatin. *J Biol Chem*. 2010;285(43):32778-86.
9. Chory EJ, Calarco JP, Hathaway NA, Bell O, Neel DS, Crabtree GR. Nucleosome Turnover Regulates Histone Methylation Patterns over the Genome. *Mol Cell*. 2019;73(1):61-72 e3.

Reviewers' comments:

Reviewer #1 (Remarks to the Author):

In the revised version of this article, the authors have addressed a number of my concerns, but many have not been adequately. Overall, in its current form, while the take-home message is very interesting, the manuscript lacks the robustness necessary to warrant publication in a journal like Nature Communications. In order to make the claims that the authors do, significant improvements would be necessary to improve certain critical aspects of the work.

Specifically, several of my previous concerns were either not addressed or not sufficiently addressed:

2. The authors' explanation about the cutoffs used is not sufficient. While it likely was not the case, one can easily get the impression that different cutoffs were arbitrarily tested for each dataset until something of significance came out. My own analysis of the TCGA data (using proteinatlas.org) provides a much more moderate effect (although still statistically significant). The authors need to provide some real justification and uniformity for the utilization of different cutoffs. For example, in the Gleason 7 samples, are the levels simply overall higher (compared to lower Gleason scores)? This would then strongly support and justify the use of different cutoffs. Also, how is the distribution of DOT1L expression? In the case that the higher Gleason scores demonstrate higher expression levels, this can easily be shown and included as support of the different cutoff utilized.

6. Data such as H3K79me2 should not be presented as Venn diagrams. This is not useful. The more useful presentation as shown in Fig. 2l gives a much better overview. The value of the presentation of data in Fig. 2m is completely unclear. At face value, one would have to interpret the findings at there being 15,926 peaks in LNCaP and 8,737 peaks in PC3 cells that appear exclusively in EPZ treated cells. Given the mechanism of H3K79 methylation deposition (directly linked to transcriptional elongation) and the block in DOT1L activity (which would prevent new addition), this cannot be explained. This significantly detracts from the credibility of the findings.

11. The answer of the authors is not consistent with the presented data. I still find the half-life experiments extremely problematic and the presentation of the data lacks credibility. The blots for the vehicle-treated samples (3g and 4j) are tremendously overexposed. Quantitation of Western blots is problematic anyway and absolutely unreliable if blots are overexposed. An independent densitometric analysis of the data shown in Fig. 4j shows that the control half-life is less than 1.5 hours (realistically, more like 30 minutes) while that with EPZ treatment is probably less. While I believe the authors' claim that the half-life of the proteins decreases with EPZ treatment, the authors' data simply do not fit together (i.e., claimed half-lives in graphs do not match Western blot data). In order for presentation in a journal of this quality, the authors must provide a more robust dataset to support their claims.

14. I appreciate that the authors have added overexpression data and individual siRNAs for HECTD4. However, individual siRNAs need to also be shown for MYCBP2 as well.

15. The authors need to add molecular weight markers to (all?) gels. Especially in the ubiquitination experiments it would be important to know at what molecular weight the unmodified AR runs and where the presumed polyubiquitinated protein runs.

19. Overall, while the authors have addressed some of my concerns with regard to the downstream enhancer on the Myc gene, I still find the evidence that this region is actually an enhancer quite weak. In the text (lines 394-397) the authors make the statement, "Thus the downstream AR-binding region is essential for MYC expression in LNCaP cells." I do not see any definitive data proving this. The authors state in their rebuttal that deletion of this enhancer is

lethal, yet they are somehow able to obtain DNA in Fig. 7i. I find it a bit unexplainable that experiments can be performed with cells with a substantial MYC knockdown (e.g., Fig. 6a), but deletion of an intergenic enhancer region close to Myc is completely lethal (although deletion of an enhancer would be expected to have a more mild effect on gene expression). Given the fact that the authors transiently transfect the gRNAs, an early time point could be used in which deletion occurs prior to lethality. In the absence of such data definitively demonstrating an effect of the inactivation of this enhancer on MYC mRNA levels (or alternative data such as dCas9-KRAB tethering), a claim that this enhancer is important for MYC expression (and thereby the entire proposed model) is simply unfounded.

Apart from these points, which have already been brought up in the first version of the paper, I have a few additional concerns:

- The manuscript contains numerous grammatical (especially capitalization) mistakes throughout.

- In lines 148-151 the authors conclude: "These results support a model wherein the functional effects of DOT1L inhibition in the AR-positive cells require events that occur after loss of H3K79 methylation such as dysregulated target gene expression in contrast to direct effects, such as modification of AR by DOT1L enzymatic activity or protein protein interaction." I cannot find any data that definitively exclude the possibility that DOT1L may be directly affecting AR and/or MYC stability e.g. through direct methylation (and thereby prevention of ubiquitination). At the very least, this statement must be revised.

- Related to #6 above. In line 165 the authors mention "...genes whose H3K79me2 was sensitive to DOT1L inhibition...". What does this mean? All H3K79 methylation is per definition sensitive to DOT1L inhibition. Judging from the legend of Suppl. Fig 2j, I'm guess the authors mean genes specifically marked with H3K79me2 in the respective cell lines. This should be clarified in the text.

- In lines 198-200 and in Fig. 3j it is apparent that the majority of AR-dependent genes actually go UP with DOT1L inhibition. The authors do a bit of hand waving to explain these effects saying this occurs "...by another transcription factor...", but when taken at face value the data clearly indicate that the majority of the AR-regulated program is not inhibited, but rather increased by EPZ treatment. This is clearly at odds with the assumption most readers would make. This does not necessarily need to detract from the major findings of the manuscript since it may very well be the case that the important tumorigenic aspects of AR biology (e.g., MYC regulation) are DOT1L dependent. However, at the very least the authors need to acknowledge this finding and more clearly describe this aspect of the biology.

- In lines 347-348 the authors state "H3K79me2 levels across the MYC gene were uniformly reduced in both LNCaP and PC 3 cells". This should be shown in the figures.

Reviewer #2 (Remarks to the Author):

The authors have substantially answered all of my questions, especially with the addition of the new PDX data and the extra cell line data. As well, in response to one of the issues raised by another reviewer, the deletion data on the putative MYC enhancer adds to the paper.

Reviewer #3 (Remarks to the Author):

In the revised manuscript by Vatapalli et al., the authors also provide new data to many of the concerns raised or authors response (AR) to other concerns. Please see below for comments on the responses provided:

1. In Figure 2m and Supp Fig 2h, what accounts for the new H3K79me2 peaks in the EPZ treated cells? What is the significance of these new peaks and do they overlap between LNCaP and PC3 cells?

AR: "We were quite intrigued by this observation of new H3K79me2 peaks that emerged upon EPZ treatment. This might be attributed to positional 'scrambling' of histones containing 'old' K79 methylation during cell replication cycles (8). It is hypothesized that H3K79 methylation is lost by replacement with new unmethylated histone and the old histones with H3K79 are displaced from their original positions (9)."

The authors have yet to respond to the last question.

2. The authors determine that there is no impact of DOT1L on AR gene regulation but are they also ruling out a physical interaction between the 2 proteins? This is particularly relevant considering the data in Fig 7 and co-localization of DOT1L and AR at the MYC enhancer.

AR: "We found evidence that AR and DOT1L interact in co-IP studies in 293T cells (Supplementary Figure 8e)."

The authors need to validate this in relevant prostate cancer cells.

3. It is not clear why it takes 8 days to observe the drop in AR (and MYC) protein? The cycloheximide data in Fig 3d and 4g, respectively suggests that these proteins turnover by 24 hours. A time course of AR and MYC expression +/- EPZ would help.

AR is fine

4. Based on Fig 3b growth curve it is suggested that DOT1L promotes androgen independent growth in LNCaP cells. It would be informative if these experiments can be done in castrate resistant, AR-positive cells (e.g. 22Rv1, C42B).

AR is fine

5. What is the status of MYC, HECTD4 and MYCBP2 in DOT1L over-expressing cells (Fig 3b)?

AR is fine

6. The data in Figs 4 and 5 suggest that DOT1L dependent MYC loss is dependent on AR activity – but this is only shown in one cell line (LNCaP) – what about castrate-resistant cells (e.g. C42B or 22Rv1)? What is the level of increase in AR and MYC protein in Fig 5d following siHECTD4/MYCBP2?

AR is fine

7. The mass spectrometry data showed 4 peptides corresponding to MYCBP2 interacting with AR. This needs co-IP validation. In addition, can the authors comment why FOXA1 or HOXB13 or other known AR interacting proteins are not on the list?

The authors need to include FOXA1 and/or HOXB13 as controls for the co-IP.

8. What is the clinical relevance of HECTD4, MYCBP2 and TRIM49 expression? How do the levels of expression for these genes correlate with DOT1L and MYC in clinical samples?

AR: "We did not find significant association of HECTD4 expression with survival. We suspect that this might be due to the effects of other unknown HECTD4 targets that might influence survival of prostate cancer cells. We did not detect significant correlation between the Ubiquitin ligases with DOT1L perhaps due to the indirect nature of the regulation."

The authors need to include this information in the manuscript in order for the reader to rule out the clinical relevance between HECTD4 and DOTL. Also, the authors have yet to respond to the last question.

9. Is the known MYC protein degradation SCFFbxw7 ubiquitin ligase or the MYC T58, S62 residues implicated in the context of DOT1L-MYC axis?

AR is fine

10. Does MYC bind to the HECTD4, MYCBP2 promoters?

AR is fine

Response to Reviewers' comments:

Reviewer #1:

In the revised version of this article, the authors have addressed a number of my concerns, but many have not been adequately. Overall, in its current form, while the take-home message is very interesting, the manuscript lacks the robustness necessary to warrant publication in a journal like Nature Communications. In order to make the claims that the authors do, significant improvements would be necessary to improve certain critical aspects of the work.

Specifically, several of my previous concerns were either not addressed or not sufficiently addressed:

2. The authors' explanation about the cutoffs used is not sufficient. While it likely was not the case, one can easily get the impression that different cutoffs were arbitrarily tested for each dataset until something of significance came out. My own analysis of the TCGA data (using proteatlas.org) provides a much more moderate effect (although still statistically significant). The authors need to provide some real justification and uniformity for the utilization of different cutoffs. For example, in the Gleason 7 samples, are the levels simply overall higher (compared to lower Gleason scores)? This would then strongly support and justify the use of different cutoffs. Also, how is the distribution of DOT1L expression? In the case that the higher Gleason scores demonstrate higher expression levels, this can easily be shown and included as support of the different cutoff utilized.

Response: We now show analysis with a uniform cutoff (25 percentile to define patients with low DOT1L expression and 75 percentile for high DOT1L expression in all three datasets). Similar to the earlier results, high DOT1L expression correlates with poor survival (**Figure 1b**). Moreover, DOT1L expression levels do increase significantly with Gleason grade (**Supplementary Figure 1h**), which as the reviewer pointed out would justify the use of the different cutoffs in our original analysis. The data analyzed with different cut-offs are now included in the supplement (**Supplementary Figure 1i**), providing the reader with a comprehensive view of the analyses.

6. Data such as H3K79me2 should not be presented as Venn diagrams. This is not useful. The more useful presentation as shown in Fig. 2l gives a much better overview. The value of the presentation of data in Fig. 2m is completely unclear. At face value, one would have to interpret the findings at there being 15,926 peaks in LNCaP and 8,737 peaks in PC3 cells that appear exclusively in EPZ treated cells. Given the mechanism of H3K79 methylation deposition (directly linked to transcriptional elongation) and the block in DOT1L activity (which would prevent new addition), this cannot be explained. This significantly detracts from the credibility of the findings.

Response: As the reviewer suggested, we have removed the Venn diagrams and present the data in the less confusing manner shown in **Figure 2l**. For **Figure 2m**, the paradoxical appearance of apparent "new peaks" in the EPZ treated samples may be due to the positional scrambling of histones that is thought to occur during replication (1, 2) which may shift where the peak calling algorithm calls a peak. We are certainly not suggesting that DOT1L inhibition leads to addition of K79me at new sites. Although a detailed study of the basis for this observation is beyond the scope of this manuscript, we feel that showing the empirical data -even if paradoxical- is more appropriate for full disclosure, while duly noting the caveats related to the interpretation.

11. The answer of the authors is not consistent with the presented data. I still find the half-life experiments extremely problematic and the presentation of the data lacks credibility. The blots for the vehicle-treated samples (3g and 4j) are tremendously overexposed. Quantitation of Western blots is problematic anyway and absolutely unreliable if blots are overexposed. An independent densitometric analysis of the data shown in Fig. 4j shows that the control half-life is less than 1.5 hours (realistically, more like 30 minutes) while that with EPZ treatment is probably less. While I believe the authors' claim that the half-life of the proteins decreases with EPZ treatment, the authors' data simply do not fit together (i.e., claimed half-life in graphs do not match Western blot data). In order for presentation in a journal of this quality, the authors must provide a more robust dataset to support their claims.

Response: We agree with the reviewer regarding the challenges of quantitation of western blots. We have performed additional experiments and endeavored to quantify bands in the "linear" range to avoid the pitfalls of overexposure as much as possible. The data are presented in new **Figures 3g and 4j**.

14. I appreciate that the authors have added overexpression data and individual siRNAs for HECTD4. However, individual siRNAs need to also be shown for MYCBP2 as well.

Response: We have now shown data with multiple MYCBP2 siRNAs (**Supplementary Fig 5d**). The knockdown results are also complemented by the overexpression data (**Figure 5c, 5d, Supplementary Fig 5h**) supporting a role for MYCBP2 in regulating AR protein.

15. The authors need to add molecular weight markers to (all?) gels. Especially in the ubiquitination experiments it would be important to know at what molecular weight the unmodified AR runs and where the presumed poly-ubiquitinated protein runs.

Response: These are now added. (**Figure 5f-h, Supplementary 5j**).

19. Overall, while the authors have addressed some of my concerns with regard to the downstream enhancer on the Myc gene, I still find the evidence that this region is actually an enhancer quite weak. In the text (lines 394-397) the authors make the statement, "Thus the downstream AR-binding region is essential for MYC expression in LNCaP cells." I do not see any definitive data proving this. The authors state in their rebuttal that deletion of this enhancer is lethal, yet they are somehow able to obtain DNA in Fig. 7i. I find it a bit unexplainable that experiments can be performed with cells with a substantial MYC knockdown (e.g., Fig. 6a), but deletion of an intergenic enhancer region close to Myc is completely lethal (although deletion of an enhancer would be expected to have a more mild effect on gene expression). Given the fact that the authors transiently transfect the gRNAs, an early time point could be used in which deletion occurs prior to lethality. In the absence of such data definitively demonstrating an effect of the inactivation of this enhancer on MYC mRNA levels (or alternative data such as dCas9-KRAB tethering), a claim that this enhancer is important for MYC expression (and thereby the entire proposed model) is simply unfounded.

Response: The reviewer raised several points that we will address as follows:

a) Evidence that the downstream AR-binding region is an enhancer. Firstly, this AR-binding region was initially discovered by the Chinnaiyan and Brown groups using ChIP-seq and ChIP-on-Chip studies in prostate cancer cells respectively (4, 5). Subsequently, Alumkal and colleagues confirmed AR binding

and H3K27 acetylation at this site (6). Moreover, others have described MYC as an AR regulated gene (7). We now show AR binding, H3K27 acetylation and H3K4me2 at this site in AR+ prostate cancer cell lines, PDX tumors and prostate cancer tissue samples (**Figure 7, Supplementary Figure 8**). Secondly, CRISPR/Cas9 deletion of this region reduced MYC expression and cell viability only in AR+ cells but not in AR- cells. More importantly, exogenous MYC expression rescued the decrease in cell viability (**Figure 7**). Thirdly, this region is K79-methylated and exhibits features of H3K79me2/3 enhancer elements or KEEs recently described in a *Nature Communications* paper (8). These authors showed that K79 methylation is required for maintaining chromatin accessibility, histone acetylation and transcription factor binding specifically at KEEs but not at non-KEE enhancers. We showed that DOT1L inhibition impairs K27 acetylation and AR binding at this region (**Figure 7e**). Our findings are consistent with this site being an AR bound KEE in AR positive cells. In addition, a recent study analyzing Pol II interactions across the genome using ChIA-PET has revealed that this downstream region of MYC interacts with the MYC promoter in AR+ VCaP cells but not in the AR- RWPE-1 or DU145 cells (9). In sum, these previous studies and our results support the contention that this downstream region is an AR-bound, K79-methylated enhancer element or KEE. We do acknowledge that further studies beyond the scope of this manuscript will still be useful to study this region in more detail.

b) Role of the downstream AR-binding region in MYC expression in LNCaP cells. The reviewer found it “bit unexplainable that experiments can be performed with cells with a substantial MYC knockdown (e.g., Fig. 6a), but deletion of an intergenic enhancer region close to Myc is completely lethal (although deletion of an enhancer would be expected to have a more mild effect on gene expression).” There is no conflict in these experiments as cells were collected at completely different time points. (Specifically, 48 h in Figure 6a and 5 days in Figure 7i). In addition, the data in Figure 6a are measurement of mRNA and not a measure of cell viability. We did note that the enhancer deletion did reduce cell viability but did not show or claim that deletion of this regions was “completely lethal”. The reduction in cell viability observed is certainly in line with expected effects when MYC expression is reduced and importantly, can be rescued by exogenous MYC. MYC expression is decreased in LNCaP cells but not PC3 cells (**Figure 7k**). This is in line with the previous study (8) showing that deletion of the KEE or K79-methylated enhancers leads to a decrease in expression of target genes (**Referee Figure 1**).

Apart from these points, which have already been brought up in the first version of the paper, I have a few additional concerns:

- The manuscript contains numerous grammatical (especially capitalization) mistakes throughout.

Response: We have proofread the manuscript and corrected the typographical errors.

- In lines 148-151 the authors conclude: “These results support a model wherein the functional effects

of DOT1L inhibition in the AR-positive cells require events that occur after loss of H3K79 methylation such as dysregulated target gene expression in contrast to direct effects, such as modification of AR by DOT1L enzymatic activity or protein-protein interaction." I cannot find any data that definitively exclude the possibility that DOT1L may be directly affecting AR and/or MYC stability e.g. through direct methylation (and thereby prevention of ubiquitination). At the very least, this statement must be revised.

Response: We agree, and do not exclude the possibility that DOT1L may directly modify AR and/or MYC. We make note of this accordingly.

- Related to #6 above. In line 165 the authors mention "...genes whose H3K79me2 was sensitive to DOT1L inhibition...". What does this mean? All H3K79 methylation is per definition sensitive to DOT1L inhibition. Judging from the legend of Suppl. Fig 2j, I'm guess the authors mean genes specifically marked with H3K79me2 in the respective cell lines. This should be clarified in the text.

Response: We thank the reviewer for catching this error. This statement is now amended.

- In lines 198-200 and in Fig. 3j it is apparent that the majority of AR-dependent genes actually go UP with DOT1L inhibition. The authors do a bit of hand waving to explain these effects saying this occurs "...by another transcription factor...", but when taken at face value the data clearly indicate that the majority of the AR-regulated program is not inhibited, but rather increased by EPZ treatment. This is clearly at odds with the assumption most readers would make. This does not necessarily need to detract from the major findings of the manuscript since it may very well be the case that the important tumorigenic aspects of AR biology (e.g., MYC regulation) are DOT1L dependent. However, at the very least the authors need to acknowledge this finding and more clearly describe this aspect of the biology.

Response: Indeed our point is precisely what the reviewer noted, that EPZ treatment led to a paradoxical increase in AR-regulated genes, which we subsequently ascribed to MYC regulation of these same target genes. We have now made this point more clear.

- In lines 347-348 the authors state "H3K79me2 levels across the MYC gene were uniformly reduced in both LNCaP and PC3 cells". This should be shown in the figures.

Response: The PC3 data were shown in the **Supplementary Fig 7a**. The H3K79me2 data for LNCaP cells were shown in **Fig 7a**.

Reviewer #3 (Remarks to the Author):

In the revised manuscript by Vatapalli et al., the authors also provide new data to many of the concerns raised or authors response (AR) to other concerns. Please see below for comments on the responses provided:

1. In Figure 2m and Supp Fig 2h, what accounts for the new H3K79me2 peaks in the EPZ treated cells? What is the significance of these new peaks and do they overlap between LNCaP and PC3 cells?

AR: "We were quite intrigued by this observation of new H3K79me2 peaks that emerged upon EPZ treatment. This might be attributed to positional 'scrambling' of histones containing 'old' K79 methylation during cell replication cycles (8). It is hypothesized that H3K79 methylation is lost by replacement with new unmethylated histone and the old histones with H3K79 are displaced from their original positions (9)."

The authors have yet to respond to the last question.

Response: Regarding the last part of the original question about the overlap of the peaks, we now show the overlap between LNCaP and PC3 for these new peaks (**Supplementary Figure 2i**).

2. The authors determine that there is no impact of DOT1L on AR gene regulation but are they also ruling out a physical interaction between the 2 proteins? This is particularly relevant considering the data in Fig 7 and co-localization of DOT1L and AR at the MYC enhancer.

AR: "We found evidence that AR and DOT1L interact in co-IP studies in 293T cells (Supplementary Figure 8e)."

The authors need to validate this in relevant prostate cancer cells.

Response: We have now performed co-IP in LNCaP DOT1L cells (**Supplementary Fig 8e**). In addition, data from a previous report (10) showed AR/DOT1L interaction.

7. The mass spectrometry data showed 4 peptides corresponding to MYCBP2 interacting with AR. This needs co-IP validation. In addition, can the authors comment why FOXA1 or HOXB13 or other known AR interacting proteins are not on the list?

The authors need to include FOXA1 and/or HOXB13 as controls for the co-IP.

Response: We have now included FOXA1 as an additional control in LNCaP co-IP to show that AR binds to FOXA1 (**Supplementary Fig 8e**).

8. What is the clinical relevance of HECTD4, MYCBP2 and TRIM49 expression? How do the levels of expression for these genes correlate with DOT1L and MYC in clinical samples?

AR: "We did not find significant association of HECTD4 expression with survival. We suspect that this might be due to the effects of other unknown HECTD4 targets that might influence survival of prostate cancer cells. We did not to detect significant correlation between the Ubiquitin ligases with DOT1L perhaps due to the indirect nature of the regulation."

The authors need to include this information in the manuscript in order for the reader to rule out the clinical relevance between HECTD4 and DOT1L. Also, the authors have yet to respond to the last question.

Response: We have now included the HECTD4 data in **Supplementary Figure 5m**. We have also shown correlation analyses between the different Ubiquitin ligases and MYC/DOT1L from 4 different datasets (**Supplementary Figure 6e**).

References:

1. Sweet SM, Li M, Thomas PM, Durbin KR, Kelleher NL. Kinetics of re-establishing H3K79 methylation marks in global human chromatin. *J Biol Chem.* 2010;285(43):32778-86.
2. Chory EJ, Calarco JP, Hathaway NA, Bell O, Neel DS, Crabtree GR. Nucleosome Turnover Regulates Histone Methylation Patterns over the Genome. *Mol Cell.* 2019;73(1):61-72 e3.
3. Guo Q, Xie J, Dang CV, Liu ET, Bishop JM. Identification of a large Myc-binding protein that contains RCC1-like repeats. *Proc Natl Acad Sci U S A.* 1998;95(16):9172-7.
4. Yu J, Yu J, Mani RS, Cao Q, Brenner CJ, Cao X, et al. An integrated network of androgen receptor, polycomb, and TMPRSS2-ERG gene fusions in prostate cancer progression. *Cancer Cell.* 2010;17(5):443-54.
5. Wang Q, Li W, Zhang Y, Yuan X, Xu K, Yu J, et al. Androgen receptor regulates a distinct transcription program in androgen-independent prostate cancer. *Cell.* 2009;138(2):245-56.
6. Gao L, Schwartzman J, Gibbs A, Lisac R, Kleinschmidt R, Wilmot B, et al. Androgen receptor promotes ligand-independent prostate cancer progression through c-Myc upregulation. *PLoS One.* 2013;8(5):e63563.
7. Bieche I, Parfait B, Tozlu S, Lidereau R, Vidaud M. Quantitation of androgen receptor gene expression in sporadic breast tumors by real-time RT-PCR: evidence that MYC is an AR-regulated gene. *Carcinogenesis.* 2001;22(9):1521-6.
8. Godfrey L, Crump NT, Thorne R, Lau IJ, Repapi E, Dimou D, et al. DOT1L inhibition reveals a distinct subset of enhancers dependent on H3K79 methylation. *Nat Commun.* 2019;10(1):2803.
9. Ramanand SG, Chen Y, Yuan J, Daescu K, Lambros M, Houlahan KE, et al. The landscape of RNA polymerase II associated chromatin interactions in prostate cancer. *J Clin Invest.* 2020.
10. Yang L, Lin C, Jin C, Yang JC, Tanasa B, Li W, et al. lncRNA-dependent mechanisms of androgen-receptor-regulated gene activation programs. *Nature.* 2013;500(7464):598-602.

REVIEWERS' COMMENTS:

Reviewer #1 (Remarks to the Author):

While I do not necessarily completely agree with the authors about to what degree they have proven that their MYC enhancer is actually responsible, I do appreciate the extensive changes that have been made to the manuscript and feel that it has become significantly stronger as a result. Thus, I would strongly support the acceptance of this manuscript for publication.

Reviewer #3 (Remarks to the Author):

The authors have sufficiently addressed my concerns.

Response to Reviewers:

Reviewer #1 (Remarks to the Author):

While I do not necessarily completely agree with the authors about to what degree they have proven that their MYC enhancer is actually responsible, I do appreciate the extensive changes that have been made to the manuscript and feel that it has become significantly stronger as a result. Thus, I would strongly support the acceptance of this manuscript for publication.

Reviewer #3 (Remarks to the Author):

The authors have sufficiently addressed my concerns.

We thank the reviewers for their comments and appreciate their support for our manuscript.